# Structural and Functional Characterization of Rv0792c from *Mycobacterium tuberculosis*: Identifying Small Molecule Inhibitor against HutC Protein

Neeraj Kumar Chauhan,[a] Anjali Anand,[a] Arun Sharma,[a] Kanika Dhiman,[b] Tannu Priya Gosain,[a] Prashant Singh,[b] Padam Singh,[a] Eshan Khan,[c] Gopinath Chattopadhyay,[d] Amit Kumar,[c] Deepak Sharma,[b] Ashish,[b] Tarun Kumar Sharma,[a]* ⓘRamandeep Singh[a]

[a]Translational Health Science and Technology Institute, Faridabad, Haryana, India
[b]Institute of Microbial Technology, Council of Scientific and Industrial Research, Chandigarh, India
[c]Department of Biosciences and Biomedical Engineering, Indian Institute of Technology Indore, Indore, India
[d]Molecular Biophysics Unit, Indian Institute of Science, Bangalore, Karnataka, India

Neeraj Kumar Chauhan and Anjali Anand contributed equally to this paper. Author order was determined on the basis of their contribution.

**ABSTRACT** In order to adapt in host tissues, microbial pathogens regulate their gene expression through a variety of transcription factors. Here, we have functionally characterized Rv0792c, a HutC homolog from *Mycobacterium tuberculosis*. In comparison to the parental strain, a strain of *M. tuberculosis* with a Rv0792c mutant was compromised for survival upon exposure to oxidative stress and infection in guinea pigs. RNA sequencing analysis revealed that Rv0792c regulates the expression of genes involved in stress adaptation and virulence of *M. tuberculosis*. Solution small-angle X-ray scattering (SAXS) data-steered model building confirmed that the C-terminal region plays a pivotal role in dimer formation. Systematic evolution of ligands by exponential enrichment (SELEX) resulted in the identification of single-strand DNA (ssDNA) aptamers that can be used as a tool to identify small-molecule inhibitors targeting Rv0792c. Using SELEX and SAXS data-based modeling, we identified residues essential for Rv0792c's aptamer binding activity. In this study, we also identified I-OMe-Tyrphostin as an inhibitor of Rv0792c's aptamer and DNA binding activity. The identified small molecule reduced the growth of intracellular *M. tuberculosis* in macrophages. The present study thus provides a detailed shape-function characterization of a HutC family of transcription factor from *M. tuberculosis*.

**IMPORTANCE** Prokaryotes encode a large number of GntR family transcription factors that are involved in various fundamental biological processes, including stress adaptation and pathogenesis. Here, we investigated the structural and functional role of Rv0792c, a HutC homolog from *M. tuberculosis*. We demonstrated that Rv0792c is essential for *M. tuberculosis* to adapt to oxidative stress and establish disease in guinea pigs. Using a systematic evolution of ligands by exponential enrichment (SELEX) approach, we identified ssDNA aptamers from a random ssDNA library that bound to Rv0792c protein. These aptamers were thoroughly characterized using biochemical and biophysical assays. Using SAXS, we determined the structural model of Rv0792c in both the presence and absence of the aptamers. Further, using a combination of SELEX and SAXS methodologies, we identified I-OMe-Tyrphostin as a potential inhibitor of Rv0792c. Here we provide a detailed functional characterization of a transcription factor belonging to the HutC family from *M. tuberculosis*.

**KEYWORDS** *Mycobacterium tuberculosis*, GntR transcription factors, HutC subfamily, bacterial pathogenesis, aptamer, SELEX, SAXS, small molecule inhibitor

**Ad Hoc Peer Reviewer** ⓘ Sandhya Visweswariah, Indian Insitute of Science

Address correspondence to Ramandeep Singh, ramandeep@thsti.res.in, or Tarun Kumar Sharma, tarun@thsti.res.in.

*Present address: Tarun Kumar Sharma, Department of Medical Biotechnology, Gujarat Biotechnology University (GBU), GIFT City, Gandhinagar, Gujarat, India.

The authors declare no conflict of interest.

[This article was published on 12 December 2022 without the present address of Tarun Kumar Sharma. The present address was added in the current version, posted on 19 December 2022.]

Mycobacterium tuberculosis, the causative agent of tuberculosis (TB), has coexisted with humans for thousands of years and is among the leading causes of mortality worldwide (1). Approximately 2.0 million people are latently infected with *M. tuberculosis* due to the ability of the pathogen to persist in host tissues (2). The incidence rates of TB are on the rise due to HIV-TB coinfections, poor patient compliance, rise of drug-resistant strains, and the failure of BCG vaccine to impart protection against adult TB (3–6). Hence, there is a compelling need for identification of novel drug targets and regimens to tackle the problem imposed by primary and latent TB infections. *M. tuberculosis* is able to sense extracellular signals and reprogram its transcriptional machinery in order to adapt to stress and persist in host tissues (7). This transcriptional reprogramming is mediated by a complex network of regulatory proteins comprising sigma factors, transcription factors, two-component systems, and serine-threonine protein kinases (8, 9). This highly coordinated regulation of gene expression in response to stress exposure is essential for *M. tuberculosis* to establish infection *in vivo*.

The GntR family of transcription factors are highly abundant in various archaeal and bacterial genomes (10, 11). This protein family was named after the gluconate operon (*gntRKPZ*) repressor of *Bacillus subtilis* and is among the most widely distributed transcription factors in prokaryotes (12, 13). These proteins harbor a highly conserved N-terminal DNA binding region and a diverse C-terminal effector binding and oligomerization domain (10, 13). GntR proteins have an extended C terminal which has not been visualized in any of the reported solved three-dimensional structures to date. The binding of effector molecules to the C-terminal domain of the GntR family of transcription factor modulates the N-terminal domain's ability to bind DNA (10, 13). Depending on the effector molecules, the GntR family of transcription regulators has been categorized into a number of subfamilies, including FadR, AraR, DevA, DasR, HutC, MocR, PlmA, and YtrA subfamilies (10, 13). Among these, FadR is the most well-characterized GntR subfamily of transcription regulators. The binding of acyl coenzyme A (acyl-CoA) to the effector domain of the FadR subfamily induces a conformational change, and DNA binding is mediated by the helix-turn-helix motif (14). In addition to regulation of gluconate metabolism, the GntR family of transcription factors has been shown to regulate microbial processes such as carbon metabolism, motility, antibiotic production, biofilm formation, and pathogenesis (15–18). The genome of *M. tuberculosis* encodes 7 homologs of the GntR family of transcription regulators. Among these, Rv0043c, Rv0165c, Rv0494, Rv0586, and Rv3060c belong to the FadR family of GntR regulators. Previously, it was shown that Rv0494 binds to fatty acyl-CoA and negatively regulates the transcription of the *kas* operon, which is involved in mycolic acid biosynthesis (19). PipR (Rv0494 homolog) in *Mycobacterium smegmatis* regulates the expression of genes involved in the utilization of piperidine and pyrrolidine (20). Rv0165c negatively regulates the *mce1* operon and is necessary for *M. tuberculosis* to persist in its host (21). Rv0586 negatively regulates the expression of the *mce2* operon and endonuclease IV (22). Rv1152, in the YtrA subfamily of GntR proteins, regulates the expression of genes required for vancomycin susceptibility in *M. tuberculosis* (23).

HutC is the second largest GntR subfamily of transcription factors and represents approximately 30% of the GntR proteins. The HutC protein from *Pseudomonas putida* represses the expression of histidine utilization genes, and its activity is regulated by urocanate binding (24). The HutC homolog from *Escherichia coli* regulates the expression of genes involved in the citric acid cycle in response to fatty acids (25). The C-terminal region of the *E. coli* HutC homolog shares structural similarity with the chorismate lyase fold, which features a buried active site behind two helix-turn-helix loops (26). In addition to histidine and fatty acids, the other known effectors for HutC protein are trehalose-6-phosphate, alkylphosphonate, and *N*-acetylglucosamine-6-phosphate (27–30). Here, we have functionally characterized Rv0792c, a transcription regulator from *M. tuberculosis*, that belongs to the HutC family. We report that Rv0792c is an autoregulatory transcription factor and is required for *M. tuberculosis* survival in oxidative stress and to establish infection in host tissues. To the best of our knowledge, this is the first report in which SELEX and solution small-angle X-ray scattering (SAXS) approaches have been combined to

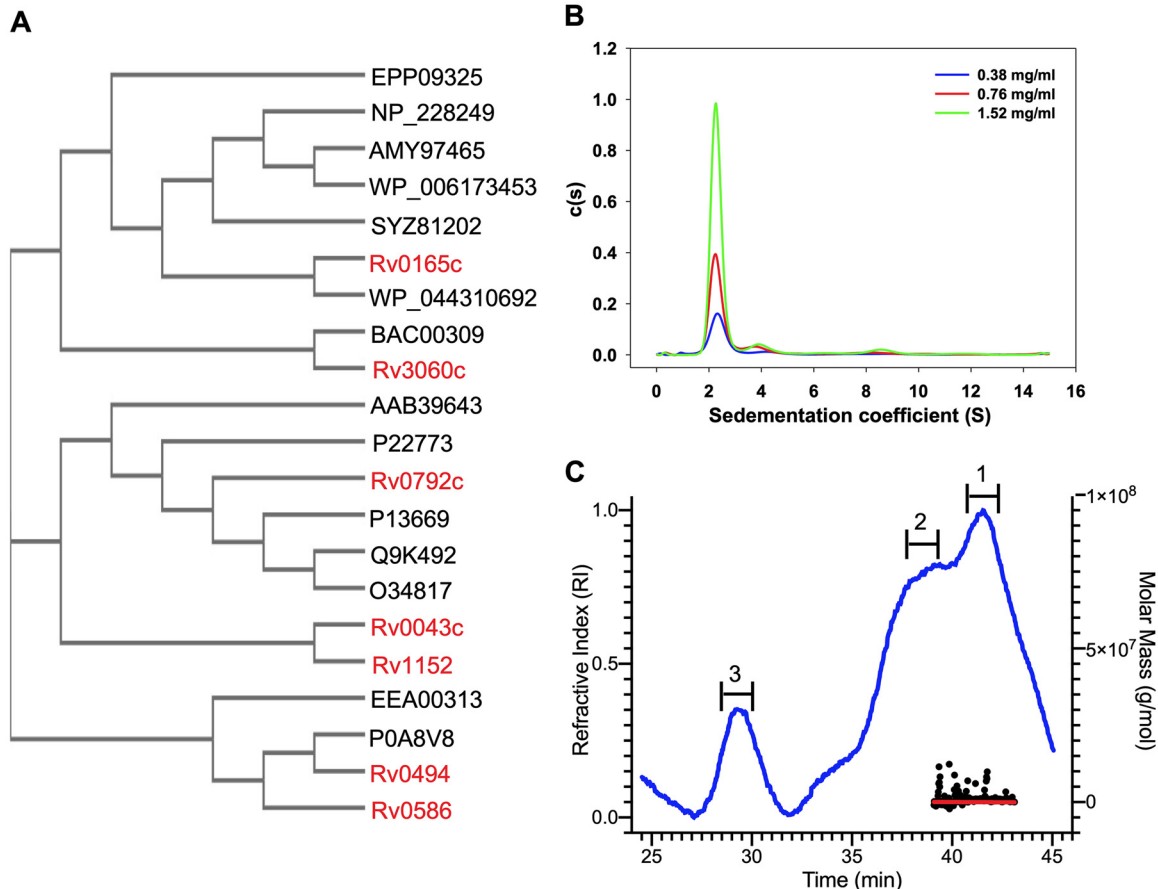

**FIG 1** (A) Rooted phylogenetic tree for GntR family of transcription factors in various prokaryotes. The bacterial species included in the phylogenetic analysis were *M. tuberculosis* (Rv0043c, Rv0165c, Rv0494, Rv0586, Rv1152, Rv0792c, and Rv3060c), *Burkholderia* sp. (EEA00313), *Pseudomonas putida* (P22773), *Escherichia coli* (P0A8V8, P13669), *Bacillus subtilis* (O34817), *Streptomyces coelicolor* (Q9K492), *Salmonella enterica* (AAB39463), *Corynebacterium glutamicum* (BAC00309), *Thermotoga maritima* (NP_228249), *Streptococcus pyogenes* (AMY97465), *Vibrio cholerae* (SYZ81202), *Pseudomonas syringae* (WP_044310692), *Brucella* sp. (WP_006173453), and *Klebsiella pneumoniae* (EPP09325). (B) Sedimentation velocity analytical ultracentrifugation experiments. Radial absorption curves for different concentrations of Rv0792c were collected at 280 nm. Scans were collected every 3 min. Sedimentation coefficient continuous distribution [$c(s)$] was plotted for the proteins at the indicated concentrations. The $c(s)$ plots showed that Rv0792c exists primarily as a dimer in its native state. (C) SEC-MALS analysis to determine oligomeric states of the purified Rv0792c. The trace for the refractive index is shown in blue. The molar mass and fit of the trace ar plotted as functions of elution time, as approximately horizontal red and black lines, respectively. The peaks analyzed for molecular weight determinations are numbered as 1, 2, and 3, where peak 1 corresponds to the dimeric form of the protein and peaks 2 and 3 correspond to higher-order aggregates.

determine (i) the structural state of Rv0792c, (ii) the aptamer binding pocket of Rv0792c, and (iii) the identification of I-OMe-Tyrphostin as a small-molecule inhibitor of Rv0792c. The identified small molecule also possessed activity against *M. tuberculosis* in THP-1 macrophages. The findings presented in this study are anticipated to open new avenues for rational design of novel small-molecule inhibitors against Rv0792c from *M. tuberculosis*.

## RESULTS

**Rv0792c from *M. tuberculosis* belongs to the HutC subfamily of GntR transcription factors.** GntR family transcription factors are highly conserved in the bacterial kingdom, and *M. tuberculosis* genome encodes seven GntR homologs (Rv0043c, Rv0165c, Rv0494, Rv0586, Rv0792c, Rv1152 and Rv3060c). Multiple-sequence alignment revealed that GntR homologs from *M. tuberculosis* shared almost identical residues in the DNA binding amino-terminal region. The effector binding region of the GntR family of transcription regulators showed little sequence identity. As shown in Fig. 1A, Rv0792c clustered with different proteins of the GntR family, such as HutC from *P. putida*, DasR from *Streptomyces coelicolor*, NagR from *B. subtilis*, PhnR from

**TABLE 1** Sedimentation coefficient values of Rv0792c obtained from analytical ultracentrifugation studies

| Rv0792c concn (mg/mL) | $S_{20,w}$ (% species) | | MW (kDa) | |
| --- | --- | --- | --- | --- |
| | Peak 1 | Peak 2 | Peak 1 | Peak 2 |
| 0.38 | 2.37 (60.5%) | 4.37 (8.2%) | 57.5 | 143.9 |
| 0.76 | 2.23 (63.8%) | 4.21 (11.6%) | 55.6 | 136.1 |
| 1.52 | 2.30 (73.6%) | 4.04 (6.7%) | 54.6 | 127.9 |

*Salmonella enterica*, and MngR from *E. coli* (10, 30, 31, 32). Multiple-sequence alignment analysis revealed that Rv0792c and other HutC homologs share an identity of ~30% among themselves (see Fig. S1 in the supplemental material). FadR homologs from *M. tuberculosis* (Rv0494, Rv0165c, Rv0586, and Rv3060c) also grouped with their respective homologs from other bacterial species (Fig. 1A). In the present study, we performed experiments to biochemically, functionally, and structurally characterize Rv0792c from *M. tuberculosis*.

**Rv0792c is a dimeric protein.** For biochemical characterization, Rv0792c was cloned in pET28b and recombinant protein was purified with an amino-terminal histidine tag. The purity of various fractions was confirmed by SDS-PAGE analysis. The purified fractions were dialyzed and concentrated, and sedimentation velocity ultracentrifugation studies were performed at various protein concentrations of 0.38 mg/mL, 0.76 mg/mL, and 1.52 mg/mL (Fig. 1B). Continuous distribution analysis of absorbance scans at different protein concentrations revealed that Rv0792c predominantly sedimented at an $s_{20,w}$ of ~2.3S, consistent with a molecular weight of ~58 kDa and thereby suggesting the protein is primarily dimeric in solution (Fig. 1B and Table 1). In addition, minor fractions of higher-order oligomeric species sedimenting at $s_{20,w}$ of ~4.3S (8 to 11%) and ~8.5S (4 to 5%), corresponding to 130 kDa and 396 kDa, respectively, were also observed (Fig. 1B). The increase in fraction of species sedimenting at ~4.3S and ~8.5S with increased protein concentration suggested formation of higher-order oligomers at relatively higher concentrations of the protein. We further probed the oligomeric status of the purified protein by size exclusion chromatography coupled with multiangle light scattering (SEC-MALS). We observed that the major peak (peak 1) for the purified Rv0792c corresponded to a homodimeric form of the protein (Fig. 1C). We also observed two additional peaks, which most likely represented higher-order aggregates (Fig. 1C).

**Rv0792c is essential for adaptation of *M. tuberculosis* upon exposure to oxidative stress.** TB infection is an outcome of *M. tuberculosis* adaptation to unfavorable environmental conditions encountered in host tissues such as low oxygen, nutritional deficiency, reactive nitrogen intermediates, oxidative stress, and acidic stress (33). Several metabolic pathways that include transcriptional regulators are essential for *M. tuberculosis* pathogenesis. The precise role of GntR homologs in *M. tuberculosis* pathogenesis has not yet been extensively deciphered. Here, we determined the function of Rv0792c in physiology, stress adaptation, and virulence of *M. tuberculosis*. We generated an Rv0792c mutant strain of *M. tuberculosis* by using temperature-sensitive mycobacteriophages (Fig. S2A). The construction of the mutant strain was confirmed by PCR and quantitative PCR (qPCR) using gene-specific primers. As shown in Fig. S2B, locus-specific primers resulted in amplification of ~1-kb and 2.1-kb bands in the case of the wild-type and mutant strain, respectively. The restoration of Rv0792c expression in the complemented strain was confirmed by qPCR (Fig. S2C). We observed no changes in growth patterns or colony morphology of the parental or Rv0792c mutant strain of *M. tuberculosis* (Fig. S2D and E).

We next compared the survival of various strains under different stress conditions *in vitro*. As shown in Fig. 2A, growth defects of ~11.0- and 20.5-fold were seen in the survival of the mutant strain compared to the parental strain after exposure to oxidative stress for 24 h and 72 h, respectively (**, $P < 0.01$) (Fig. 2A). This growth defect associated with the mutant strain upon exposure to oxidative stress was restored in the complemented strain (Fig. 2A). We observed that both wild-type and mutant

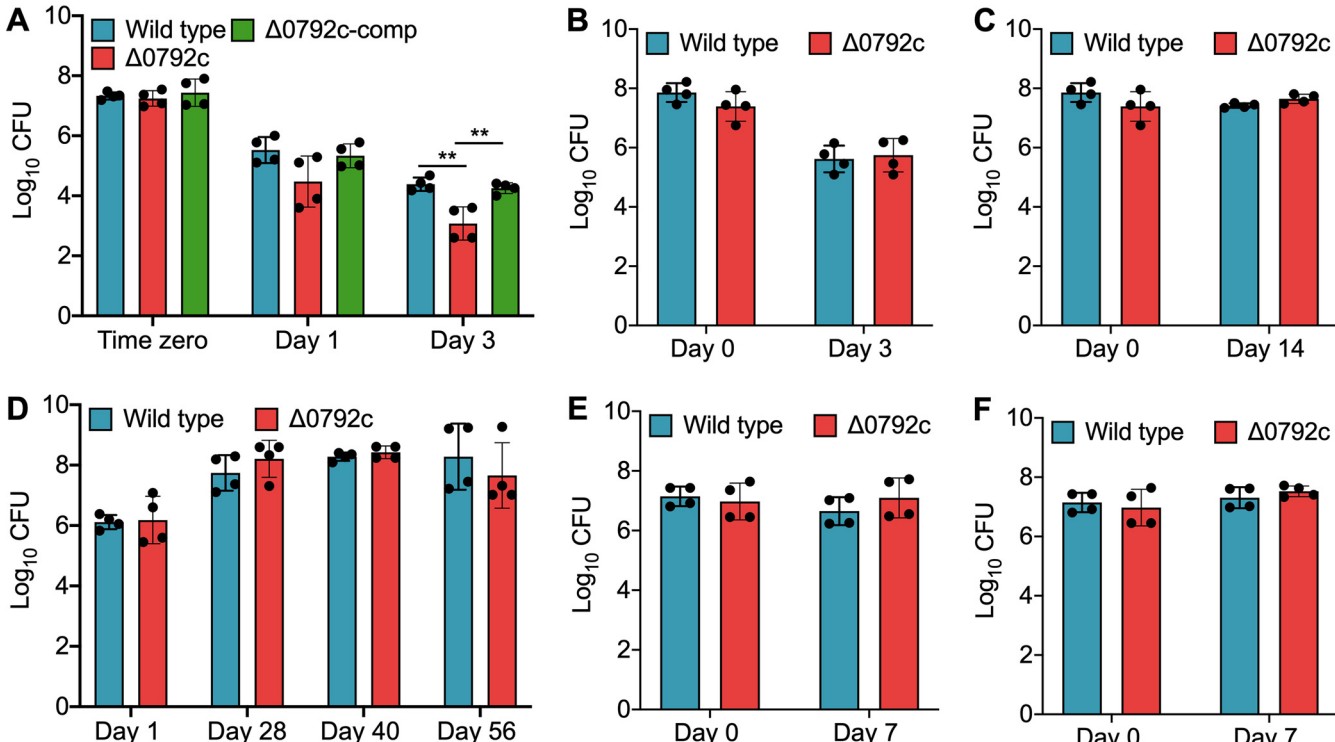

**FIG 2** Effect of deletion of Rv0792c on survival of *M. tuberculosis* under different stress conditions *in vitro*. For *in vitro* stress experiments, early-log-phase cultures of various strains were exposed to either oxidative stress (5 mM $H_2O_2$) (A) or nitrosative (5 mM $NaNO_2$) (B) or nutritional stress (C), or low oxygen (D), or acidic pH 4.5 (E) or acidic pH 5.2 (F). The data shown in these panels are means $\pm$ standard deviations (SD) of the $\log_{10}$(CFU) obtained from two independent experiments performed in duplicates. Statistically significant differences were observed for the indicated groups (one-way ANOVA using Tukey's method; **, $P < 0.01$).

strains were susceptible to comparable levels after exposure to other stress conditions tested in this study (Fig. 2B to F). Since GntRs have been shown to be involved in the biofilm formation of bacterial pathogens, we also determined the role of Rv0792c in *M. tuberculosis* biofilm formation *in vitro* (34–37). We observed that parental and Rv0792c mutant strains of *M. tuberculosis* were comparable in their abilities to form biofilms *in vitro* (data not shown). In order to understand the role of Rv0792c in drug tolerance, we next determined the sensitivity of various strains to drugs with different mechanisms of action. Figure S2F clearly shows that deletion of Rv0792c had no effect on *M. tuberculosis* susceptibility after 14 days of exposure to the indicated drugs. In concordance, both strains displayed comparable $MIC_{99}$ values of 0.39 $\mu$M, 2 nM, 0.78 $\mu$M, and 3.125 $\mu$M against isoniazid, rifampicin, levofloxacin, and ethambutol, respectively (data not shown). Taken together, we demonstrated that Rv0792c is important for the ability of *M. tuberculosis* to adapt to oxidative stress *in vitro*.

**Deletion of Rv0792c impaired the ability of *M. tuberculosis* to cause infection in guinea pigs.** Next, we determined the ability of Rv0792c to contribute to *M. tuberculosis* pathogenesis using the guinea pig model of infection. For animal experiments, guinea pigs were infected with various strains, and aerosol infection resulted in implantation of ~50 to 100 bacilli in lungs at day 1 postinfection. We found discrete multiple lesions in lung tissues of animals infected with the parental or complemented strain. The number of these lesions was significantly reduced in lung tissues from mutant strain-infected guinea pigs at both 28 and 56 days postinfection (Fig. 3A). We observed no differences in the lung tissue weights of guinea pigs infected with the various strains (data not shown). As shown in Fig. 3B, at 4 weeks postinfection, ~550.0-fold significantly higher bacterial numbers were observed in lungs of wild-type strain-infected guinea pigs, in comparison to mutant strain-infected guinea pigs (****, $P < 0.0001$). In agreement with these findings, guinea pigs infected with the wild-type strain had splenic bacillary

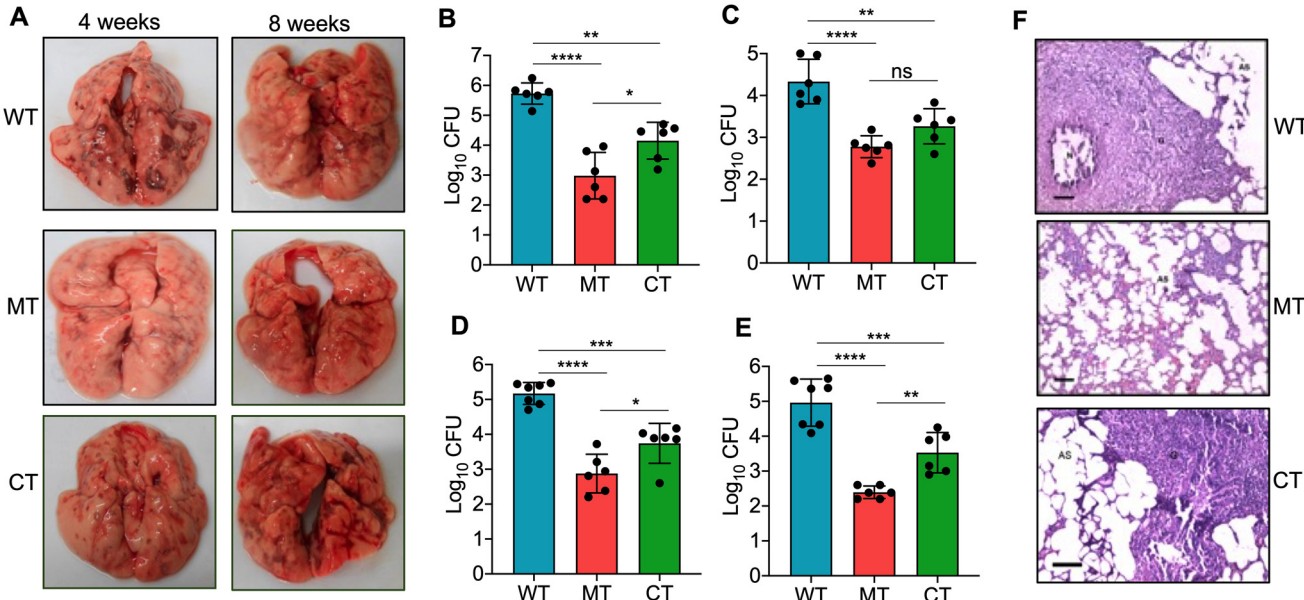

**FIG 3** Rv0792c is essential for *M. tuberculosis* to establish infection in the host. (A) Gross pathological evaluation of tissue damage of lung tissues from guinea pigs infected with either wild-type, Rv0792c mutant, or Rv0792c-complemented strains at 4 weeks or 8 weeks postinfection. The images are representative of those obtained from 6 or 7 animals per time point for each strain. (B to E) The lung and splenic bacillary loads were determined in aerosol-infected guinea pigs at both 4 weeks (B and C) and 8 weeks (D and E) postinfection. The data presented in this panel are means ± SD of $\log_{10}$(CFU) obtained from 6 or 7 animals per time point for each strain. Statistically significant differences were observed for the indicated groups (one-way ANOVA using Tukey's method; *, $P < 0.05$; **, $P < 0.01$; ***, $P < 0.001$; ****, $P < 0.0001$). (F) Tissue sections from animals 8 weeks postinfection were stained with hematoxylin and eosin to determine the extent of tissue damage. Scale bar, 100 $\mu$m. Magnification, $\times100$. The images shown are representative of those obtained from 6 or 7 animals per time point for each strain. The abbreviations used in the panel are as follows: WT, wild-type strain (H37Rv); MT, Rv0792c mutant strain; CT, Rv0792c complemented strain.

burdens that were ~36.0-fold higher than those infected with the mutant strain (Fig. 3C) (****, $P < 0.0001$). Further, the lungs and splenic bacillary loads were reduced by ~200.0- and ~370.0-fold, respectively, in mutant strain-infected guinea pigs, in comparison to wild-type-infected guinea pigs at 56 days postinfection (Fig. 3D and E) (****, $P < 0.001$). At both 4 and 8 weeks postinfection, complementation with Rv0792c partially restored the growth defect associated with the mutant strain (Fig. 3B to E) (*, $P < 0.05$; **, $P < 0.01$). Concordantly, minimal tissue involvement was observed in hematoxylin and eosin (H&E)-stained sections from guinea pigs infected with the mutant strain at 8 weeks postinfection (Fig. 3F). Granuloma formation was seen in sections from animals infected with either wild-type or complemented strains (Fig. 3F). Taken together, our findings demonstrated that Rv0792c is required for *M. tuberculosis* to establish infection in host tissues but not for *in vitro* growth.

**Effect of deletion of Rv0792c on the transcriptional profile of *M. tuberculosis*.** The observed growth defect of the Rv0792c mutant strain in guinea pigs raised the possibility that it regulates the expression of genes essential for *M. tuberculosis* virulence. In order to characterize the Rv0792c regulon, RNA sequencing (RNA-seq) experiments were performed using total RNA isolated from mid-log-phase cultures of the wild type and the mutant strain as described in Materials and Methods. We observed that the majority of genes were expressed to comparable levels in both strains. Using a cutoff value of a >2.0-fold change and a $P$ value of <0.05, transcriptome analysis revealed that a total of 197 genes were differentially expressed in the mutant strain in comparison to the wild-type strain (Fig. 4A). Among these, the levels of 108 and 89 transcripts were increased and decreased, respectively, in the mutant strain (Fig. 4A; Table S3). These differentially expressed genes (DEGs) were further characterized based on their annotations in Mycobrowser (https://mycobrowser.epfl.ch/). We noticed that most of the DEGs were either conserved hypothetical proteins or involved in processes such as cell wall synthesis or intermediary metabolism (Fig. 4B). The transcript levels of

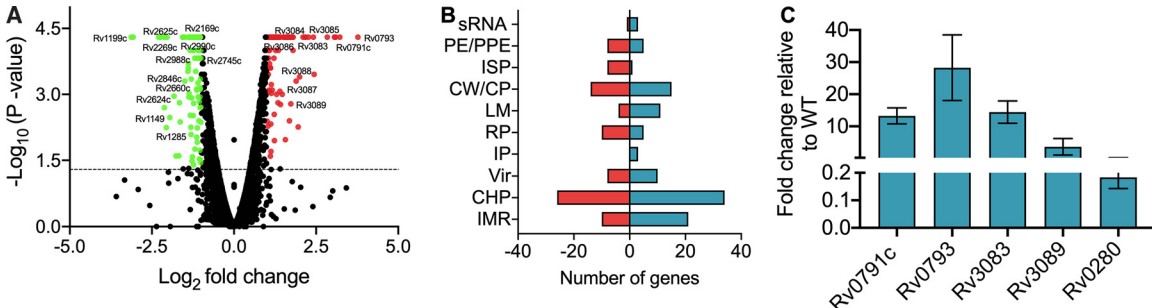

**FIG 4** Differential expression of genes in the Rv0792c mutant strain. (A) Volcano plot illustrating the gene expression comparison between wild-type and mutant strains, where individual genes are represented as single dots with the $-\log_{10}$ P value on the y axis and the $\log_2$ fold change on the x axis. Selected subsets of significant genes are shown with their gene names as the labels. The data shown in this panel were obtained from three replicate samples. The upregulated and downregulated genes in the mutant strain are shown by red and green dots, respectively. Black dots represents genes whose expression was not statistically significant between the wild-type and mutant strains. (B) The functional annotation of differentially expressed genes between parental and Rv0792c strain is shown. These annotations were generated using TB database (https://mycobrowser.epfl.ch). The abbreviations used in the panel are as follows: IMR, intermediary metabolism and respiration; CHP, conserved hypothetical protein; Vir, virulence, detoxification, and adaptation; IP, information pathways; RP, regulatory proteins; LM, lipid metabolism; CW/CP, cell wall and cell processes; ISP, insertion sequences and phages; PE/PPE, proline-glutamic acid (PE)/proline-proline-glutamic acid (PPE); sRNA, small RNA. (C) The relative expression of differentially expressed transcripts was quantified after normalization to levels of housekeeping gene *sigA*. The gene IDs are shown on the x axis and mean ± SD fold changes obtained from three independent experiments are shown on the y axis.

genes encoding proteins such as Rv0383c, Rv1094 (*desA2*), Rv1285 (*cysD*), Rv1350 (*fabG2*), Rv2166c, Rv2846c, Rv2988c, and Rv3139, which are essential for *M. tuberculosis* growth *in vitro*, were downregulated in the mutant strain (Table S3). RNA-seq analysis revealed that the transcripts of genes upregulated under low-oxygen conditions (e.g., Rv2624c, Rv2625c, Rv3126c, Rv0572c, and Rv1734c) or nutrient-limiting conditions (e.g., Rv1149, Rv1285, Rv1929c, Rv2169c, Rv2269c, Rv2660c, and Rv2745c) were also reduced in the mutant strain (Table S3) (38, 39). Among the upregulated genes, the mutant strain had higher transcript levels of the neighboring genes for Rv0792c and *mymA* (Table S3) (40, 41). A subset of these differentially expressed genes in RNA-seq experiments was also assessed by qPCR. As expected, the expression patterns obtained by qPCR were similar to those obtained from RNA-seq data (Fig. 4C). These observations suggested that Rv0792c regulates gene expression and this transcriptional reprogramming is required for *M. tuberculosis* to adapt and survive in host tissues.

**Generation of Rv0792c binding aptamer through SELEX.** We next performed SELEX experiments to find DNA aptamers as possible tools to identify an epitope(s) which binds small-molecule inhibitors against Rv0792c. The SELEX was performed using an 80-nucleotide-long random ssDNA library. To diversify the sequences of the aptamer library, SELEX binding experiments were performed using an error-prone *Taq* DNA polymerase (42). Prior to successive SELEX rounds, the double-stranded (ds) PCR products were converted to single-stranded forms, using previously reported methods (42). After 6 rounds of SELEX, the enrichment of Rv0792c-specific binders was determined by aptamer-linked immobilized sorbent assay (ALISA). We observed saturation in the binding of the aptamer pool to Rv0792c after round 3 of SELEX. In comparison to the enriched population (round 1 to round 6), we observed negligible binding of Rv0792c with the naive library (nonenriched) (data not shown). Therefore, aptamer binders from round 6 of SELEX enrichment were cloned in the pTZ57R/T vector. The plasmid DNA from 17 randomly picked transformants was isolated and subjected to DNA sequencing. The sequences of aptamer candidates were then further analyzed using ClustalW and BioEdit software (Fig. 5A; the dots show nucleotide conservation at a particular position). We also identified sequence conservation logos by using the online tool MEME Suite 5.4.1 (https://meme-suite.org/meme/). As shown in Table S4, logos 1, 2, and 4 were among the most represented logos.

Phylogenetic analysis revealed two preponderant groups among aptamer sequences

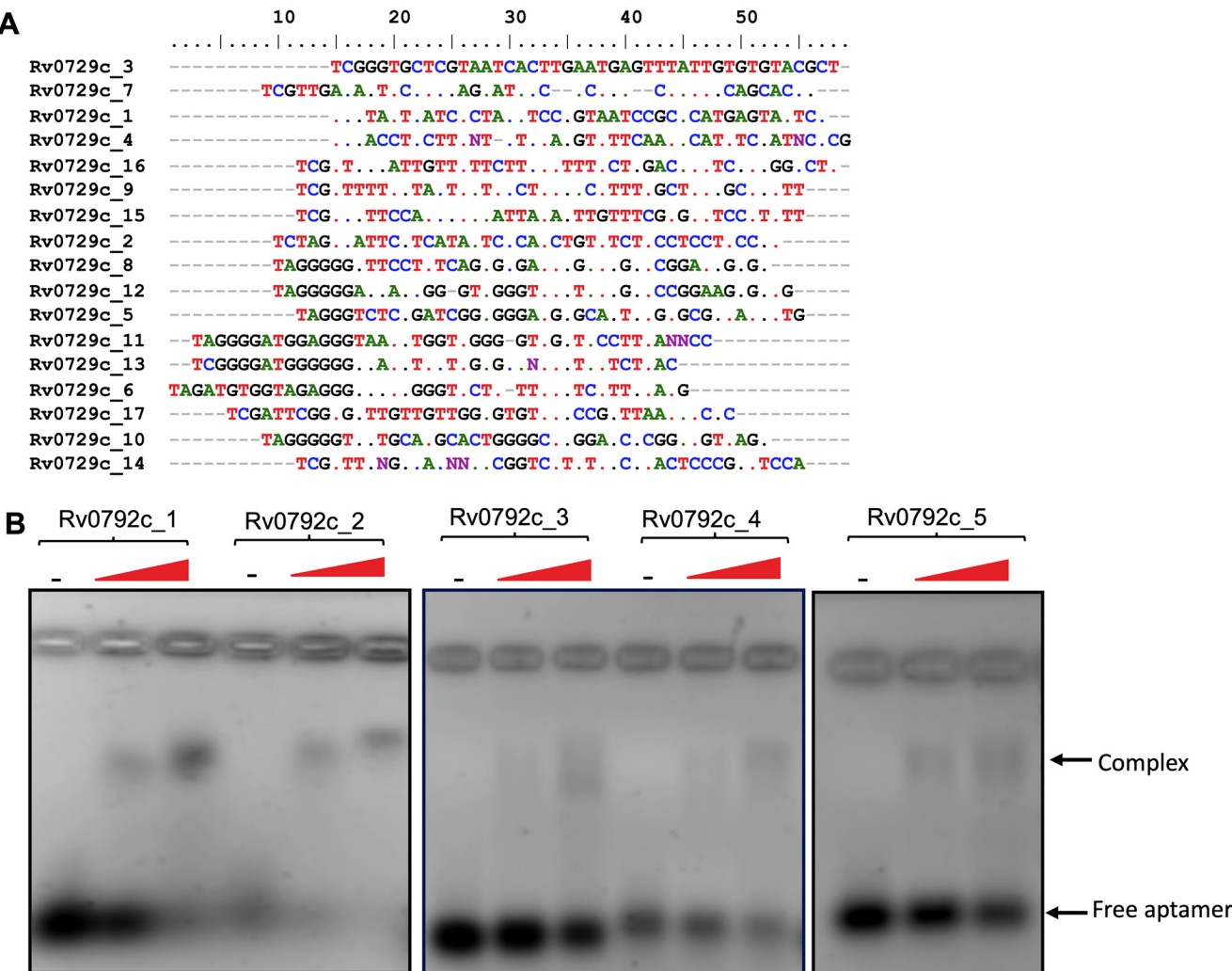

**FIG 5** Selection and characterization of Rv0792c binding aptamers. (A) Multiple-sequence alignment of Rv0792c binding aptamer sequences. (B) EMSA of selected aptamer candidates (100 pmol) with increasing concentrations of purified Rv0792c. The data in panel B are representative of two independent experiments.

(Fig. S3A). Group 1 was the largest and contained 11 aptamer candidates, while group 2 had 6 aptamers (Fig. S3A). These SELEX-derived aptamer candidates showed higher distributions of G and T residues over A and C residues in the base fraction analysis (Fig. S3B). Based on primary sequence similarity, 5 representative aptamer candidates were selected for further studies. Interestingly, Rv0729c_1, Rv0729c_2, and Rv0729c_3 showed substantial similarities with the known DNA sequences for the GntR family of bacterial transcription factors (Fig. S3C) (13).

Next, to confirm the binding of aptamers with Rv0792c, electrophoretic mobility shift assays (EMSAs) using a panel of selected aptamer candidates (Rv0729c_1, _2, _3, _4, and _5) were performed. As shown in Fig. 5B, we observed that these aptamers interacted with Rv0792c at varied strengths. Based on these findings, we selected the three best aptamer candidates, namely, Rv0792c_1, Rv0792c_2, and Rv0792c_5, for further biochemical and functional characterizations of Rv0792c. The aptamers Rv0792c_1, Rv0792c_2, and Rv0792c_5 were highly specific in their binding to Rv0792c, as no interaction was found with other *M. tuberculosis* proteins, such as HspX, GlcB, and MPT-64 (Fig. 6A). The N-terminal DNA binding region of the GntR proteins is highly conserved (14, 43). Multiple-sequence alignment analysis revealed that residues important for DNA binding, Arg49 and Gly80 of Rv0792c, which correspond to Arg35 and Gly66 of FadR

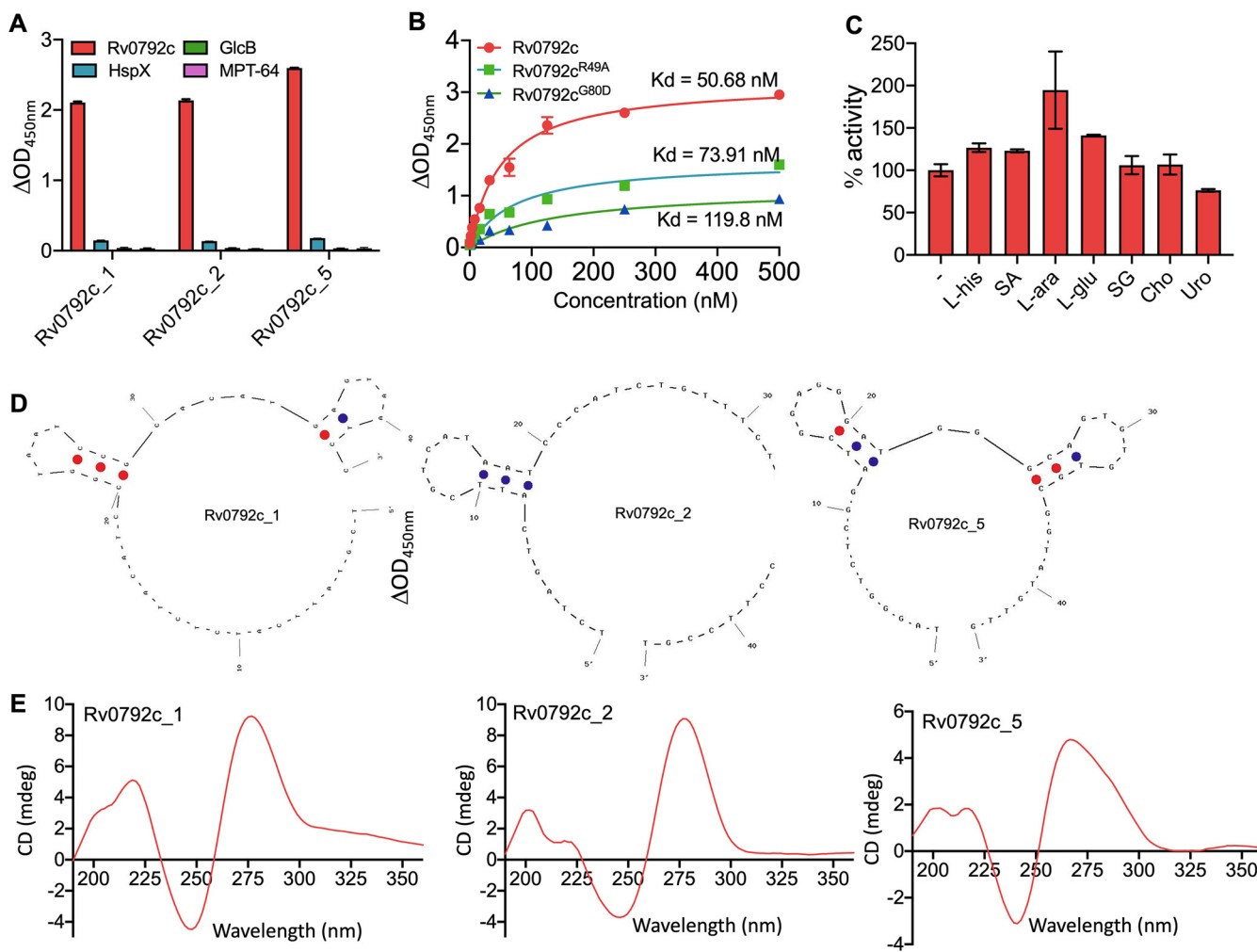

**FIG 6** Binding and structural studies of selected aptamer candidates. (A) Binding of 1 $\mu$M selected aptamer candidates (Rv0792c_1, Rv0792c_2, and Rv0792c_5) with either Rv0792c, GlcB, HspX, or MPT-64. The data shown in this panel are means $\pm$ SD of data obtained from three experiments. (B) Apparent dissociation constant curve derived through nonlinear regression representing binding affinity ($K_d$) of Rv0792c_2 for the wild type and Rv0792c[R49A] and Rv0792c[G80D]. The data shown in panel B are means $\pm$ SD obtained from two experiments. (C) Effects of various ligands on aptamer binding to Rv0792c protein. L-his, SA, L-ara, L-glu, SG, Cho, and Uro represent L-histidine, stearic acid, L-arabinose, L-glucose, sodium glyoxylate, cholesterol, and urocanic acid, respectively. The data shown in this panel are means $\pm$ SD obtained from two experiments. (D) UNA fold predicated secondary structure of selected aptamer candidates. (E) CD spectrum of Rv0792c_1, Rv0792c_2, and Rv0792c_5 aptamers. CD spectra indicate the typical B-type stem-loop structure of aptamers.

proteins, were conserved. We next determined the dissociation constant for binding of the aptamer Rv0729c_2 with wild-type and mutant proteins, and fitting of data was performed using the nonlinear regression method. The wild-type protein showed the strongest binding, as expected, with a $K_d$ value of ~51 nM. In comparison, Rv0792c[R49A] and Rv0792c[G80D] bind to Rv0729c_2 with $K_d$ values of ~74 nM and ~120 nM, respectively (Fig. 6B). Since Rv0792c is a member of the HutC family of GntR regulators, we next examined whether L-histidine or urocanic acid (an intermediate in L-histidine catabolism) acted as effectors and altered the ability of Rv0792c to bind DNA (44–47). In order to identify Rv0792c effector molecules, ALISA was performed in the presence of various ligands. As shown in Fig. 6C, ALISA activity results indicated that Rv0792c aptamer binding ability did not increase in the presence of various ligands except for L-arabinose (Fig. 6C). An ~2.0-fold increase was seen in the presence of L-arabinose (Fig. 6C). We found that the aptamer binding ability of Rv0792c was not affected in the presence of L-histidine and urocanic acid (Fig. 6C). These observations suggest that L-arabinose might act as an effector molecule for the DNA binding activity of Rv0792c.

**Secondary structure and circular dichroism spectral analysis.** In order to determine the structures of the selected aptamers, they were subjected to UNA fold analysis (Integrated DNA Technologies). The secondary structures of selected aptamer candidates (Rv0792c_1, Rv0792c_2, and Rv0792c_5) that were selected showed a typical hairpin (stem-loop) like structure (Fig. 6D). Next, we performed circular dichroism (CD) studies to determine their conformation. As shown in Fig. 6E, CD spectra of Rv0792c_1, Rv0792c_2, and Rv0792c_5 revealed a prominent positive and negative peak around 275 to 280 nm and 240 nm, respectively. The observed spectra were in concordance with the spectra obtained for stem-loop-like DNA (48, 49). However, the observed differences in the amplitude could be attributed to the variations in the sequences of these aptamers.

**SAXS data-based structural model of dimeric Rv0792c in the presence or absence of aptamers.** Currently, there is no high-resolution structure of Rv0792c; however, in a previous report modeling studies revealed the dimeric nature of the protein (50). In the present study, we acquired SAXS data to build a structural model in solution and determine the aptamer binding regions for Rv0792c. SAXS data were collected for Rv0792c at a concentration of 3.2 mg/mL, as shown in Fig. S4A. The double-logarithm mode of presentation confirmed a lack of aggregation or interparticulate effect in the protein sample (51). The inset in Fig. S4A shows the Guinier region, which considers the globular scattering nature and linear fit to the analysis, confirming the monodisperse profile of the sample. Guinier analysis suggested the particle size to be characterized by a radius of gyration ($R_g$) of about 3.3 nm (Table S2b). Indirect Fourier transformation of the data provided a frequency distribution of pairwise interatomic vectors, which further provided an estimate of the maximum linear dimension ($D_{max}$) and $R_g$ of 12.5 and 3.31 nm, respectively (Fig. S4B). Molecular mass estimation from different Bayesian models applied on the experimental SAXS data suggested that the mass of the scattering particles was ~63.2 ± 5.7 kDa (mean ± standard deviation [SD]), supporting a dimeric state of association in solution (the theoretical mass of the monomer is 32.5 kDa). The dimeric state of the protein obtained using SAXS was consistent with the previously modeled structure of Rv0792c and our area under the curve (AUC) and SEC-MALS results (50).

As described in Materials and Methods, a dummy residue model best representing the scattering shape of Rv0792c in solution was restored by averaging 10 independent models and is presented in transparent map format with variation among models reflected in wire format (Fig. 7A and Fig. S5A). A normalized spatial disposition (NSD) value of 0.93 supported the similarity of the 10 models solved and averaged for Rv0792c using SAXS data (Table S2b). In order to compare the SAXS-based envelope with the structural model of Rv0792c, a sequence-based homology model was searched. The best sequence identity of 18.14% was observed between residues 53 to 285 of Rv0792c with the solved structure for protein lin2111 from *Listeria innocua* Clip11262 (PDB deposition 3EDP; unpublished structure). We observed that most templates were similar in terms of the fold with a predicted association state of dimer. As stated in Materials and Methods, the missing 52 and 16 residues from the amino and carboxy terminus, respectively, were modeled, their predominant conformation was oriented, and they were subsequently attached to this central structural model of the Rv0792c dimeric structure. Inertial axes of this structure were superimposed with those of the SAXS-based model for Rv0792c, and similarities in the profile were visually judged in the orthogonal views (Fig. 7A and Fig. S5A). Furthermore, a $\chi^2$ value of 1.3 between the theoretical SAXS profile of the residue-level model of Rv0792c and the experimental data supported a similarity between the two models in three dimensions (Table S2b). The zoomed-in image in Fig. 7A highlights that the two C-terminal extensions of chains bound to each other, thus contributing to additional stabilization of the dimeric entity.

Further, in order to perceive local and relative flexibility embedded in the computed structure of dimeric Rv0792c, low-frequency normal modes accessible to the protein were calculated (Fig. S5B). The collective modes indicated that the amino-terminal domain moved in a synchronized mode independent of the central $\beta$-barrel-

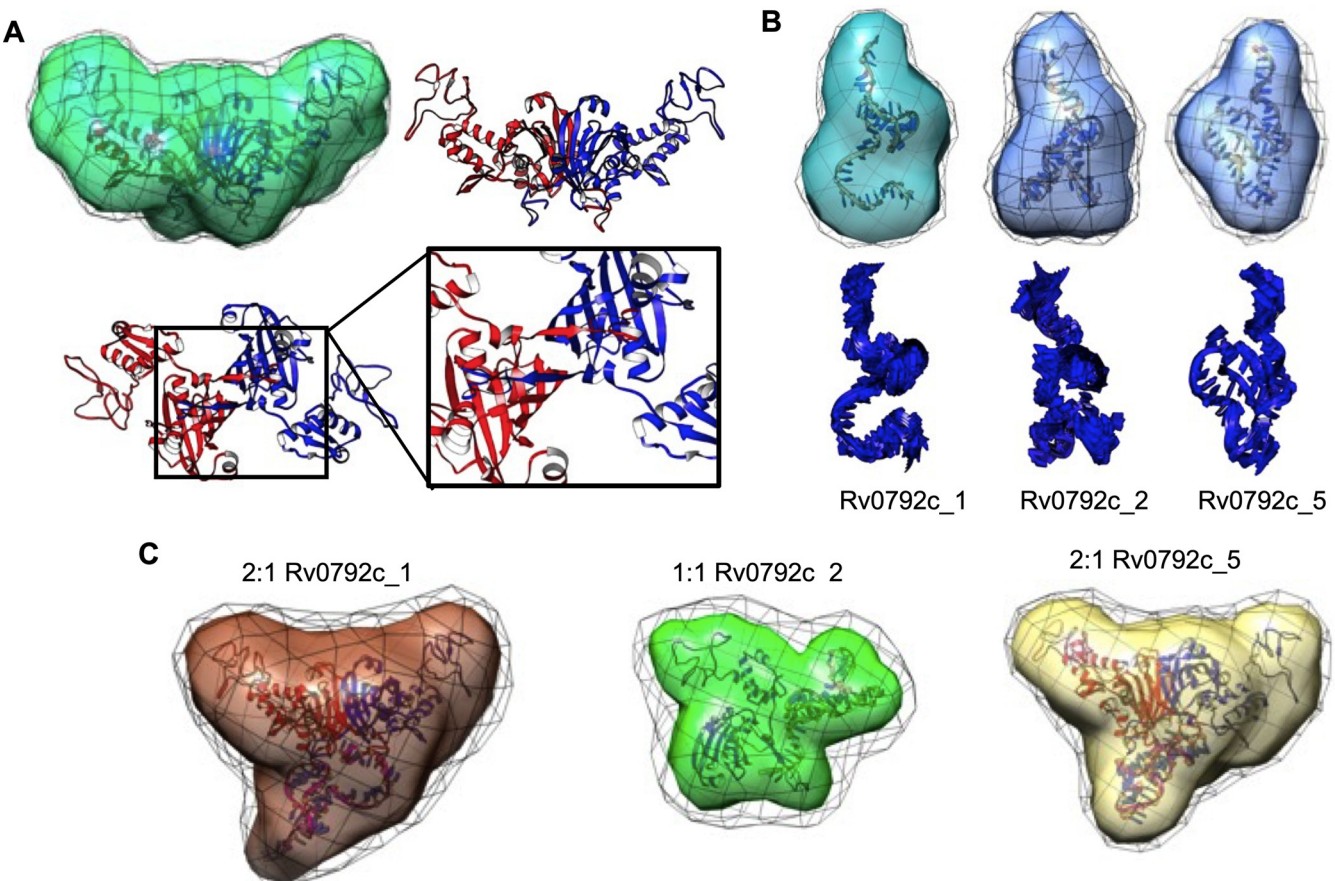

**FIG 7** SAXS-based structural models of unliganded protein Rv0792c, its binding aptamers, and their complexes. (A) SAXS data-based envelope of Rv0792c dimer (green map) inertially aligned on the structural model of the protein (blue and red ribbons). The green map shows the common shape in 10 solutions, and black mesh indicates the variations in them. Other panels show the C-terminal tail of each chain bound to the other stabilizing the dimeric association. (B, upper) SAXS-based dummy residue models of the unliganded aptamers Rv0792c_1, Rv0792c_2, and Rv0792c_5 as a molecular map. The corresponding models were aligned with the inertial axes of the model from energy optimization (shown as a ribbon). (Lower) Most collective normal-mode frequency calculated for the energy-optimized model of aptamer to reflect the inherent motion to the model. (C) SAXS data-based models of ternary and binary complexes of Rv0792c_1, Rv0792c_2, and Rv0792c_5 and protein Rv0792c are shown as molecular maps. The black mesh indicates the variations in the models solved for that complex. Inside each map is an energy-minimized model of the complex generated by SAXS data-based selected results from docking of a low-energy conformation of aptamer on protein, shown as ribbons. Note that binding of Rv0792c_2 to protein led to a binary complex, while other aptamers formed stable ternary complexes.

type dimeric contact. The carboxy-terminal tail of the proteins also moved up and down the interacting $\beta$-barrel and linker connecting the barrel and amino-terminal domain of the other chain in the dimer. These theoretical analyses imply that the carboxy-terminal ends of the dimer remain attached to each other. Next, using SAXS data analysis, solution shape parameters, association state, and structure of Rv0792c binding aptamers were determined in their unliganded state (Fig. 7B). A double-log profile of SAXS data from aptamers confirmed lack of any aggregation or interparticulate effect in the samples (Fig. S4C). Guinier analysis for globular scattering profiles are shown in the inset, and linearity of the fits in low-$q$ range further validated the monodisperse nature of the aptamers. The parameters deduced for the predominant scattering shapes of aptamers are listed in Table S2b. In summary, aptamers had $R_g$ and $D_{max}$ values in the range of 1.8 to 2.1 nm and 7.1 to 8.6 nm, respectively. For all aptamers, the calculated P($r$) indicated a "tailing" at higher $r$ values, suggesting flexible ends about the core shape (Fig. S4D). Using SAXS data profiles, the estimated molecular masses were in the range of 12.9 to 13.5 kDa, clearly supporting a monomeric state of these ssDNA molecules in solution. Their dummy residue models solved within SAXS data-based constraints are shown in Fig. 7B. As mentioned in Materials and Methods, considering monomeric status, predominant low-energy conformations of the ssDNA

aptamers were calculated in an implicit dielectric of 80 (representing water), and the best-resembling conformation was selected using the lowest $\chi^2$ value between the calculated SAXS profile for the conformation and experimental data. These models for aptamers are shown overlaid on the SAXS data-based model and alone in Fig. 7B. The additional views are shown in Fig. S6A, B, and C. Relative to Rv0792c, NSD values for aptamers in the range of 0.5 to 0.7 indicated the differential nature of the 10 models solved for the three aptamers (Table S2b). This implied a relatively higher inherent disorder in the unliganded aptamers as monomers. Similar disorders were observed in the higher-terminal motions in the residue-level models computed for these aptamers. Pertinently, it also explained the extended nature of their computed P(r) curves.

Having characterized the Rv0792c protein as adopting a dimeric state in solution and that all the binding aptamers are monomers, the next set of SAXS data was acquired on molar mixtures of the protein and individual aptamers (the dimeric:monomeric state ratio was computed for Rv0792c and ssDNA) (Fig. S4E and F). It is important to state here that the concentration of the molecules was higher than the estimated binding constant of the protein and DNA molecules, supporting a higher order of binding between available molecules and a scope of none or few unliganded molecules in samples used for SAXS data collection. Double-log plot and Guinier analyses of the Rv0792c_1, Rv0792c_2, and Rv0792c_5 aptamers supported that the scattering molecules did not aggregate or underwent an interparticulate effect upon mixing (Fig. S4E). While mixtures of Rv0792c with aptamers Rv0792c_5 and Rv0792c_1 showed $D_{max}$ and $R_g$ values of 12.5 and 3.4 to 3.5 nm, respectively, the complex of GntR and Rv0792c_2 aptamer adopted decreased $D_{max}$ and $R_g$ values of 9.8 and 3.1 nm, respectively (Table S2b). The lower-dimension complex with Rv0792c_2 correlated with the SAXS-based molecular mass prediction, which indicated a mass of ~44 kDa for this complex and ~75 kDa for samples with Rv0792c and aptamers Rv0792c_5 and Rv0792c_1. These results indicated that Rv0792c_5 and Rv0792c_1 bind to dimeric protein and do not alter the association state of Rv0792c into monomers. In contrast, binding of Rv0792c_2 aptamer induced dissociation of dimeric proteins into monomers. The observed differences with $D_{max}$ and $R_g$ values and P(r) profiles between the Rv0792c-aptamer complex, unliganded Rv0792c, and aptamers supported that protein and aptamer molecules were bound to each other during data collection. Shapes restored for these scattering species showed higher NSD values than unliganded aptamers.

As mentioned in Materials and Methods, results from SAXS data analysis revealed that Rv0792c_5 and Rv0792c_1 form a 2:1 complex with Rv0792c, but binding of Rv0792c_2 dissociates the Rv0792c dimer into monomers. Accordingly, the low-energy structures of aptamers were docked on Rv0792c dimers for Rv0792c_5 and Rv0792c_1 aptamers and on monomer for the Rv0792c_2 aptamer to obtain models for their complexes. For the latter, an approximation was made that entailed no large shape change theoretically detaching monomer from dimer of Rv0792c. Different poses of docked aptamers on Rv0792c were filtered to correlate with the shape solved for the complexes (Fig. 7C and Fig. S7). The models selected for Rv0792c: aptamer complexes indicated that aptamers bound to the C-terminal portion of Rv0792c (Fig. 8A). Energy minimization of the residue-level models of complexes obtained from docking indicated that while Rv0792c_5 and Rv0792c_1 aptamers coalesced with the dimer interface, Rv0792c_2 aptamer induced opening of the C-tail latch of Rv0792c. Probably, this last event in the case of Rv0792c_2 weakens the protein-protein interaction between Rv0792c and leads to eventual formation of a 1:1 complex. In summary, all aptamers remain monomeric in the presence or absence of Rv0792c and bind to its C-tail region, and some interaction extends to the stretch encompassing PRG (residues 40 to 42) of the Rv0792c protein (Fig. 8A, box in the zoomed image in the lower panel).

**Computational docking to identify small-molecule inhibitors for Rv0792c.** As seen from shape restoration and docking data, all screened aptamers were bound to the C-terminal dimerizing segment of the Rv0792c protein. Additionally, binding of Rv0792c_2 induced dissociation of the Rv0792c dimer. Presuming that this segment is

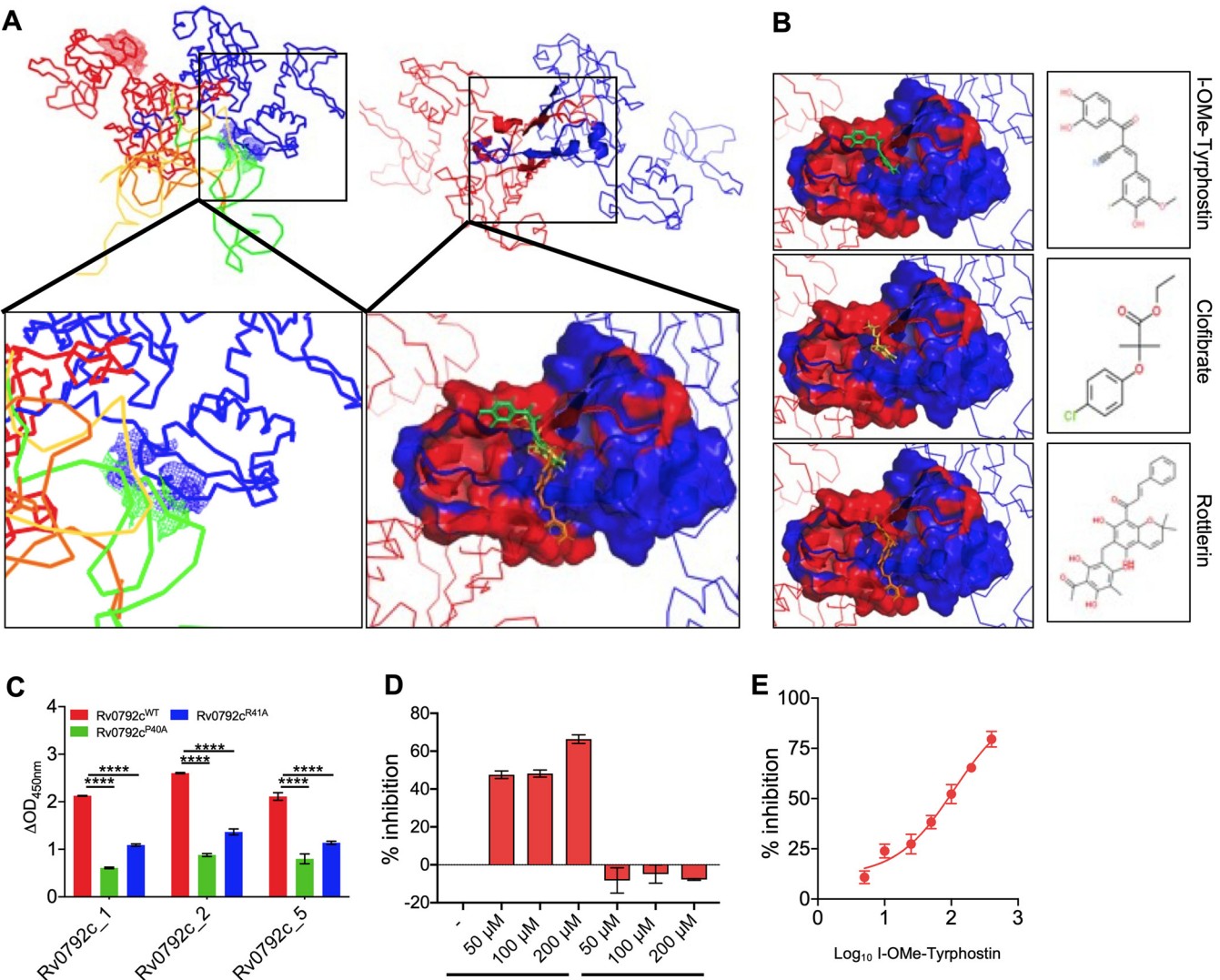

**FIG 8** Identification of I-OMe-Tyrphostin as a small-molecule inhibitor of Rv0792c. (Upper) Ribbon representations of the similar binding of the three aptamers in space near the C-terminal dimerization region (left) and the segment which was common to binding of all aptamers (right). (Lower left) The zoomed-in panel showing how aptamers binding to the C-terminal part extended the interaction to the functionally critical PRG motif in one chain of the protein (shown as blue mesh). (Lower right) The zoomed-in image shows the docking of three best hits on the surface common to binding of aptamers. The receptor is shown in surface map mode with coloring of the chain of the dimer. The molecules are shown as sticks. (B) The right three panels show the docked pose of individual molecules with their chemical structures in inset. From top to bottom, the hit molecules were the following: (2E)-2-(3,4-dihydroxybenzoyl)-3-(4-hydroxy-3-iodo-5-methoxy phenyl) acrylonitrile (I-OMe-Tyrphostin); 2-(4-chlorophenoxy)-2-methylpropionic acid ethyl ester (clofibrate); (2E)-1-[6-(3-acetyl-2,4,6-trihydroxy-5-methylbenzyl)-5,7-dihydroxy-2,2-dimethyl-2H-chromen-8-yl]-3-phenyl-2-propen-1-one (rottlerin). (C) Binding of selected aptamer candidates (Rv0792c_1, Rv0792c_2, and Rv0792c_5) with wild type, Rv0792c$^{P40A}$, or Rv0792c$^{R41A}$. The data are means ± SD from 3 experiments. The statistically significant differences were observed for the indicated groups (one-way ANOVA using Dunnett method; ****, $P < 0.0001$). (D) The aptamer binding activity of Rv0792c was determined by ALISA in the presence of I-OMe-Tyrphostin and clofibrate at 50 $\mu$M, 100 $\mu$M, and 200 $\mu$M. (E) Binding of Rv0792c with aptamer was determined in the presence of different concentrations of I-OMe-Tyrphostin, as described in Materials and Methods. The data shown in panels D and E were obtained from 2 experiments.

key to structural organization of Rv0792c and small molecules capable of binding this segment may alter the native functioning of this protein, we used the aptamer binding segments to screen for molecules which could bind to this protein. Figure 8B shows the defined aptamer binding region and three best hits in their lowest-energy pose with the receptor. These molecules were I-OMe-Tyrphostin, clofibrate, and rottlerin, in decreasing order of their relative docking scores. These small molecules differed structurally from the virtual screening-identified potential drug leads against Rv0792c (50). Next, we performed ALISA experiments to investigate whether (i) the PRG motif is required for aptamer binding to Rv0792c and (ii) whether the small molecules identified

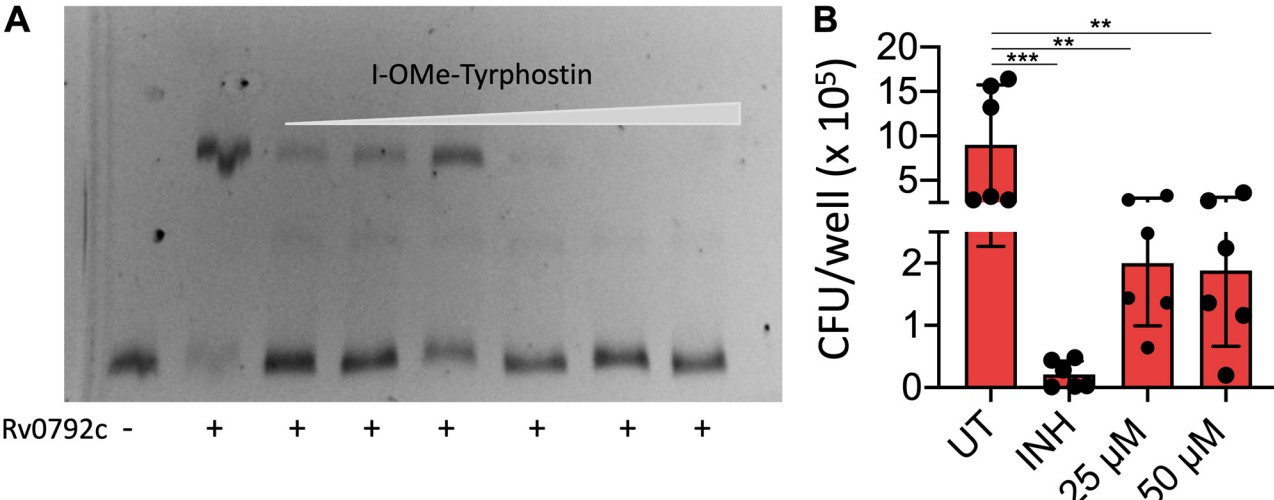

**FIG 9** I-O-Me-Tyrphostin inhibits DNA binding ability of Rv0792c (A) and intracellular growth of *M. tuberculosis* in macrophages (B). (A) EMSAs of (His)$_6$-Rv0792c protein with the Cy5-labeled promoter region of Rv0792c in the presence of varying concentration of I-O-Me-Tyrphostin (0.78 $\mu$M to 25 $\mu$M). The protein was preincubated with drugs at room temperature for 15 min before the addition of labeled DNA. The reaction mixtures were electrophoresed on 6% polyacrylamide gels. The data shown in this panel are representative of two independent experiments. (B) THP-1 macrophages were infected with *M. tuberculosis* followed by treatment with INH or I-O-Me-Tyrphostin for 96 h. For CFU enumeration, the macrophages were lysed and 100 $\mu$L of 10-fold serial dilutions were plated on MB7H11 medium. The data shown are means $\pm$ SD obtained of for the log$_{10}$ CFU per milliter, obtained from two independent experiments performed in triplicate wells. The statistically significant differences were observed for the indicated groups (one-way ANOVA using Dunnett method, **, $P < 0.01$; ***, $P < 0.001$).

from computational docking could inhibit the binding of Rv0792c_2 aptamer to Rv0792c. In order to determine the role of the PRG motif in binding to these aptamers, Rv0792c harboring Pro$^{40}$-Ala$^{40}$ and Arg$^{41}$-Ala$^{41}$ mutations was cloned, expressed, and purified as a His$_6$-tagged protein. In concordance with SAXS-based modeling results, we observed that mutation of either proline 40 or arginine 41 to alanine reduced the aptamer binding ability of Rv0792c (Fig. 8C). Of the three compounds, we were only able to procure two compounds, and these were evaluated for their ability to inhibit Rv0792c aptamer binding activity. I-OMe-Tyrphostin was able to inhibit the binding of aptamer to Rv0792c protein by ~60% at 200 $\mu$M concentration (Fig. 8D). We noticed no inhibition of aptamer binding in the presence of clofibrate, even at 200 $\mu$M concentration (Fig. 8D). As shown in Fig. 8E, I-O-Me-Tyrphostin had a concentration-dependent ability to inhibit Rv0792c aptamer binding activity and displayed a 50% inhibitory concentration (IC$_{50}$) of ~109 nM.

**I-O-Me-Tyrphostin inhibits DNA binding activity of Rv0792c and growth of *M. tuberculosis* in macrophages.** It has been previously reported that the GntR family of transcription factors bind to their native promoters and autoregulate their expression (52, 53). Therefore, we performed EMSAs to study the binding of purified Rv0792c with its native promoter. As shown in Fig. 9A, the purified protein was able to bind to the Cy5-labeled promoter. Clear retardation was seen in the mobility of labeled DNA in the presence of the purified protein (Fig. 9A). These observations suggested that, similar to other GntR homologs, Rv0792c binds to its own promoter and probably autoregulates its expression (54, 55). Next, we also determined whether I-O-Me-Tyrphostin was able to inhibit the DNA binding activity of Rv0792c. We performed EMSA using Cy5-labeled promoter fragment in the presence of different concentrations of I-O-Me-Tyrphostin. As shown in Fig. 9A, we noticed that preincubation of Rv0792c with I-O-Me-Tyrphostin decreased the promoter binding activity of Rv0792c in a concentration-dependent manner. A macrophage infection model is a useful tool to determine efficacies of drugs against intracellular *M. tuberculosis*. We next determined the activity of I-O-Me-Tyrphostin against intracellular *M. tuberculosis*-infected THP-1 macrophages. We observed that I-O-Me-Tyrphostin was able to inhibit the growth of intracellular *M. tuberculosis*. In comparison to a dimethyl sulfoxide control, exposure to 25 $\mu$M or 50 $\mu$M I-O-Me-Tyrphostin inhibited the growth of intracellular *M.*

*tuberculosis* by 4.5-fold (Fig. 9B) (**, $P < 0.01$). As shown in Fig. 9B, exposure to isoniazid (INH) reduced the growth of intracellular *M. tuberculosis* by 45-fold (***, $P < 0.001$). This study demonstrated for the first time the necessity of the *M. tuberculosis* HutC protein Rv0792c for the establishment of infection in host tissues. We have also reported here novel aptamers which bind to the dimerizing segment of the protein. Using this information, we were able to identify an FDA-approved drug that targets Rv0792c and inhibits the growth of intracellular *M. tuberculosis*.

## DISCUSSION

Transcriptional regulation has been shown to be essential for adaptation of various bacterial pathogens upon exposure to unfavorable and harsh environmental conditions. The *M. tuberculosis* genome encodes several transcription regulators that have been demonstrated to be essential for its growth *in vitro* or *in vivo*. GntR family transcription factors are widespread among prokaryotes and are involved in various processes, including (i) carbon metabolism, (ii) motility, (iii) antibiotic production, (iv) drug tolerance, (v) biofilm formation, and (vi) virulence and pathogenesis (15–18, 23). Despite the presence of GntR homologs in the genome of *M. tuberculosis*, their biological functions have not been established extensively (24–26). In the present study, we used different approaches to characterize Rv0792c protein (GntR homolog belonging to the HutC subfamily) from *M. tuberculosis*. Using sedimentation ultracentrifugation and SEC-MALS, we determined unambiguously that Rv0792c exists as a dimer in solution, as reported earlier for this protein and other GntR homologs (50, 54). In order to investigate the role of the HutC homolog in physiology, stress adaptation, and virulence, Rv0792c mutant strain was generated using temperature-sensitive mycobacteriophages. We noticed that the colony morphology, growth pattern, and biofilm formation of the mutant and parental strain were comparable. The mutant strain was compromised for survival upon exposure to oxidative stress. However, no differences were observed between wild-type and mutant strains upon exposure to other stress conditions or various drugs with a different mechanism of action. Notably, we observed significantly higher lung and splenic bacillary loads in guinea pigs infected with the wild-type strain, in comparison to those infected with the mutant strain. In agreement, less tissue damage was seen in H&E-stained sections from guinea pigs infected with the mutant strain. The granuloma formation was seen in sections of guinea pigs infected with the wild-type or complemented strains. These observations implied that Rv0792c plays a key role in *M. tuberculosis* virulence and is essential for establishing infection in host tissues. The comparison of transcriptomic profiles of wild-type and mutant strains revealed several important genes linked to *M. tuberculosis* virulence and survival, such as *leuC* (Rv2987c), *desA2* (Rv1094), *efpA* (Rv2846c), and *clgR* (Rv2745c), which were differentially expressed. The transcripts of several genes belonging to either PE/PPE or toxin-antitoxin modules or lipid metabolism were also downregulated in the mutant strains. We also observed that the transcript levels of genes encoding Rv0793 (the gene neighboring Rv0792c) and *mymA* operon (shown to be upregulated under acidic conditions) were increased in the mutant strain (41, 55). Interestingly, the transcript levels of genes present downstream of Rv0792c, such as Rv0791c and Rv0790c, were also increased in the mutant strain. These findings suggested an alternative transcriptional start site just upstream of Rv0791c. Taken together, these observations indicated that Rv0792c regulates the expression of a "subset" of genes that enables the bacteria to adapt and persist in host tissues.

A SELEX strategy was employed to search for novel ssDNA aptamers capable of tightly binding Rv0792c. Our extended aim was to utilize the aptamer binding information to screen for small molecules which may efficiently bind Rv0792c and act as inhibitors of its function. The direct interactions between Rv0792c and SELEX-derived DNA aptamers were confirmed using EMSA and ALISA. Among the identified aptamers, three DNA aptamer candidates (Rv0792c_1, Rv0792c_2, and Rv0792c_5) showed good binding to Rv0792c, albeit with varied affinities. This difference in the affinity of the DNA-protein complex evinced the differential rate of aptamer-target complex association and dissociation (48, 56). Notably, the primary sequences of Rv0729c_1,

Rv0729c_2s and Rv0729c_3 DNA aptamer candidates, designed against Rv0792c, a HutC protein, showed high similarity to DNA binding sequences of FadR subfamily transcription factors (13). We observed that the level of similarity was much higher in the cases of Rv0729c_1 and Rv0729c_2 compared to Rv0729c_5. This pattern clearly indicated the possible role of these nucleotides in providing affinity to bind the transcription factor of the GntR family. Sequence alignment studies revealed that the highly conserved helix-turn-helix DNA binding domain of Rv0792c spans amino acids between residues 21 and 86. Interestingly, the binding of all three aptamers was reduced by the substitution of glycine 80 to aspartic acid and of arginine 49 to alanine in the helix-turn-helix motif. This was possibly due to two reasons: (i) aptamer selection was performed with the wild-type protein and (ii) these residues (glycine 80 and arginine 49) play a critical role in maintaining the protein structure where aptamer binds (43). In the GntR family, the conformation of the DNA binding motif is altered by binding of effector molecules. The binding of fatty acyl-CoA negatively regulates DNA binding activity of FadR in *E. coli* (57). Similarly, DNA binding of AraR in *B. subtilis* is repressed by binding of arabinose to the effector binding domain (58). The activity of the HutC subfamily has also been reported to be regulated by *N*-acetylglucosamine and urocanic acid (28, 47). We observed that inclusion of L-histidine and urocanic acid did not affect the DNA binding ability of Rv0792c. However, arabinose increased the DNA binding ability of Rv0792c by ~2.0-fold, which suggested that L-arabinose might be the effector molecule for Rv0792c. Nevertheless, regulation of Rv0792c might still be fine-tuned by this specific ligand interaction and/or by some unknown ligand that needs to be investigated in the near future.

Furthermore, to gain insight into the complexes of aptamers bound to Rv0792c, SAXS data analysis and molecular modeling were utilized. Analysis of the unliganded protein and aptamers showed that in solution their association states were predominantly dimer and monomer, respectively. The dimeric status of Rv0792c correlated well with the AUC and SEC-MALS data, which showed presence of minor higher-order associated species too. Interestingly, mixing of aptamers to dimeric Rv0792c showed that while one molecule of Rv0792c_1 aptamer and Rv0792c_5 aptamer bound to one dimer of Rv0792c, binding of Rv0792c_2 aptamer led to dissociation of the Rv0792c dimer into monomer. *In silico* molecular modeling steered and selected within constraints from experimental SAXS data provided a key insight that Rv0792c dimerizes across its C terminal, and the extended C-tail wraps around each other chain that provides additional stability to the dimer. The dimeric status or even association architecture is not novel to the GntR family of proteins, but the unique wrapping up of C-tails on each other chain opens up queries on its functional relevance and possible uniqueness in this family of regulators. Solution scattering data supported that the aptamers remained predominantly monomer. Interestingly, the structural analysis revealed that the exposed side of the dimeric Rv0792c was also the interaction site of the three aptamers that were identified from the SELEX study. Taken together, SAXS data provided the insight that binding of Rv0792c_2 aptamer induces rearrangement(s), which leads to dissociation of the dimer of Rv0792c protein.

Taking a cue from the poses of aptamers on the Rv0792c dimer, we considered using the interacting residues in the protein to screen for small molecules which may even compete with the binding of aptamers and DNA. The two molecules from the identified top hits were experimentally evaluated in our aptamer binding assays, and we observed that I-OMe-Tyrphostin was able to inhibit binding of Rv0792c_2 aptamer to Rv0792c. The identified small molecule also inhibited the ability of Rv0792c to binds to its native promoter in a dose-dependent manner. It is worth mentioning here that Rv0792c_2 binds with the highest affinity to Rv0792c, so it can be safely extrapolated that the Tyrphostin analog may also competitively inhibit binding of other two aptamers. This molecule, I-OMe-Tyrphostin, and its analogs have been assayed before for their potential as epigenetic regulators (59, 60). No specific correlations have been made with *M. tuberculosis*, except for a recent study which identified new inhibitors for

the Pup proteasome system in *M. tuberculosis* (61). They reported that I-OMe-Tyrphostin and Tyrphostin inhibited Dop, a depupylase from *M. tuberculosis*. We also determined the ability of I-O-Me-Tyrphostin to inhibit the growth of intracellular *M. tuberculosis*. We observed that the lead molecule was able to inhibit the growth of *M. tuberculosis* in macrophages. In conclusion, we have (i) delineated the role and contribution of HutC protein in *M. tuberculosis* physiology, stress tolerance, and pathogenesis and (ii) also identified a small-molecule inhibitor against Rv0792c, an *in vivo* essential transcription factor. Future experiments should involve chromatin immunoprecipitation sequencing experiments to identify the direct binding regions of Rv0792c and screening of libraries to identify more-potent small-molecule inhibitors. The detailed functional characterization performed in the present study paves the way for design of more-potent small-molecule inhibitors against the HutC protein from *M. tuberculosis*.

## MATERIALS AND METHODS

**Bacterial strains, media, and growth conditions.** The strains, plasmids, and primers used in the study are shown in Table S1 in the supplemental material. Various *E. coli* strains were cultured in Luria-Bertani broth medium. *M. tuberculosis* $H_{37}Rv$ was used as a parental strain in this study. Unless mentioned, culturing of *M. tuberculosis* strains was performed in Middlebrook 7H9 and Middlebrook 7H11 media as previously described (62). When required, antibiotics were added at the following concentrations: ampicillin, 100 $\mu$g/mL for *E. coli*; kanamycin, 25 $\mu$g/mL for both *E. coli* and mycobacteria; hygromycin, 150 $\mu$g/mL for *E. coli* and 50 $\mu$g/mL for mycobacteria; chloramphenicol, 34 $\mu$g/mL for *E. coli*. Biofilm experiments were performed in slightly modified Sauton's medium containing ferric ammonium citrate (50 mg/liter), $MgSO_4 \cdot 7H_2O$ (0.05 g/liter); $ZnSO_4$ (0.01 mg/liter); $K_2HPO_4$ (1.0 g/liter), $CaCl_2$ (0.05 g/liter); asparagine (0.5 g/liter); $Na_2HPO_4$ (2.5 g/liter), and Tyloxapol (0.05%). $MIC_{99}$ determination experiments were carried out using the microdilution method in the presence of different drugs, as described previously (63).

**Protein expression and purification.** DNA fragments coding for either wild-type Rv0792c or mutant proteins (Rv0792c$^{R49A}$, Rv0792c$^{G80D}$, Rv0792c$^{R41A}$, or Rv0792c$^{P40A}$) were cloned in the isopropyl-$\beta$-D-thiogalactopyranoside (IPTG)-inducible prokaryotic expression vector pET28b. For protein expression and purification, various constructs were transformed into the *E. coli* BL21-CodonPlus strain. The expression of recombinant proteins was induced by the addition of 1.0 mM IPTG at 18°C with shaking at 200 rpm for 12 to 16 h. The induced cultures were harvested by centrifugation at 6,000 × *g*, pellets were resuspended, and clarified lysates were prepared by sonication in 1× phosphate-buffered saline (PBS). The recombinant protein from the clarified lysates was purified using $Ni^{2+}$-nitrilotriacetic acid (NTA) chromatography. The purified fractions were analyzed by SDS-PAGE, pooled, dialyzed, concentrated, and stored at −80°C in buffer (10 mM Tris [pH 7.4], 100 mM NaCl, 5% glycerol) until further use. HspX and GlcB proteins were purified as described previously (64).

**Sedimentation velocity and SEC-MALS experiments.** Analytical ultracentrifuge experiments were performed in a Beckman Optima XL-I analytical ultracentrifuge equipped with an absorbance-based detection system. The two-sector charcoal centerpiece with a 1.2-cm path length and built with sapphire windows was used for the study. The cell was filled with 390 $\mu$L of protein sample at a concentration of 0.38 mg/mL, 0.76 mg/mL, or 1.52 mg/mL in buffer A (25 mM HEPES [pH 7.2], 400 mM NaCl, 10% glycerol). The reference cell was filled with 400 $\mu$L of buffer A. Radial scans were collected at 40,000 rpm for 12 to 14 h with an interval of 3 min at 280 nm. The data analysis was performed with the SedFit analysis program for continuous distribution(s) models based on the Lamm equation. The Sednterp program was used to measure various parameters, such as buffer density ($\rho$ = 1.01660), buffer viscosity ($\eta$ = 0.01057 P), and partial specific volume (υ = 0.73677) of the proteins. Standard sedimentation coefficients of samples were reported as $S_{20,w}$, referring sedimentation at 20°C. For SEC-MALS studies, the oligomeric state of purified Rv0792c was analyzed on a Superdex-200 analytical gel filtration column (GE Healthcare) equilibrated in storage buffer (10 mM Tris, 100 mM NaCl; pH 7.4) with inline UV (Shimadzu), MALS (mini DAWN TREOS; Wyatt Technology), and refractive index detectors (Waters 24614) for molecular weight analysis. For each run, 200 $\mu$g of purified protein was injected at a flow rate of 0.4 $\mu$L/mL, and UV, MALS, and RI data were collected and analyzed using ASTRA software (Wyatt Technology) as described previously (65).

**Generation of the Rv0792c mutant and complemented strain of *M. tuberculosis*.** In order to determine the function of Rv0792c in the *M. tuberculosis* physiology and pathogenesis, the mutant strain was constructed using temperature-sensitive mycobacteriophages as per standard protocols (66). The replacement of the Rv0792c coding region with the hygromycin resistance gene in the mutant strain was confirmed by PCR and qPCR using locus-specific primers. For the generation of a complemented strain, Rv0792c was amplified along with its native promoter and cloned into an integrative vector, pMV306K. The electrocompetent cells of the Rv0792c mutant strain were electroporated with the recombinant plasmid pMV306K-Rv0792c, and the transformants were selected on Middlebrook 7H11 medium supplemented with kanamycin and hygromycin.

**Stress experiments.** In order to assess the contribution of Rv0792c in stress adaptation, early-log-phase cultures (optical density at 600 nm [$OD_{600}$] of ~0.2) of various strains were subjected to different

stress conditions as previously described (62, 67). For drug tolerance experiments, mid-log-phase cultures ($OD_{600}$, ~1.0) of various strains were exposed to either isoniazid, levofloxacin, or rifampin at 10 $\mu$g/mL, 10 $\mu$g/mL, and 0.4 $\mu$g/mL, respectively. For bacterial load enumeration, 10-fold serial dilutions were prepared and 100-$\mu$L aliquots were plated on Middlebrook 7H11 medium at 37°C for 3 to 4 weeks.

**Animal experiments.** For animal experiments, pathogen-free female guinea pigs (Hartley strain, 250 to 300 g) were purchased from Lala Lajpat Rai University of Veterinary and Animal Sciences, Hisar, India. The strains were grown to mid-log phase and washed twice with 1× PBS, and single-cell suspensions were prepared for aerosol infection. The guinea pigs were infected using a Glas-Col aerosol chamber with single-cell suspensions of the various strains that resulted in implantation of 50 to 100 bacilli at day 1 postinfection. The extent of disease progression in guinea pigs was determined by both CFU counts and histopathology analysis as described previously (62).

**Ethics statement.** The infection experiments were supervised as per Committee for the Purpose and Supervision of Experiments on Animals guidelines and performed at the Infectious Disease Research Facility, Translational Health Science and Technology Institute, New Delhi, India. The animal experiments were approved by the institutional animal ethics committee of Translational Health Science and Technology Institute.

**RNA-seq analysis.** For RNA-seq analysis, total RNA was isolated from mid-log phase cultures ($OD_{600}$, ~0.8) of parental and Rv0792c mutant strains using the TRIzol method as described previously (62). The purified RNA was shipped to Aggrigenome Labs, Pvt., Ltd. (India) for sequencing. The preparation of library, RNA-seq, and data analysis were performed as previously described (68). The transcripts showing differential expression of at least 2.0-fold with a *P* value of <0.05 were considered significant. Quantitative PCR was performed to validate the identified DEGs obtained from RNA-seq analysis. For qPCR studies, DNase I-treated RNA was subjected to cDNA synthesis using SuperScript III reverse transcriptase. The synthesized cDNA was diluted 1:5, and qPCR was performed using gene-specific primers and SYBR green mix. The expression of genes of interest was normalized to the transcript levels of *sigA* (a housekeeping gene) and was quantified as previously described (62).

**Aptamer selection against Rv0792c.** Aptamer selection against Rv0792c was performed using the SELEX method as previously described (69). Briefly, to identify unique aptamer sequences specific for Rv0792c, an aptamer library (2,000 picomoles) in selection buffer (SB; 10 mM Tris [pH 7.5], 10 mM $MgCl_2$, 50 mM KCl, 25 mM NaCl) was allowed to bind nitrocellulose membrane (NCM) to remove the NCM binding species. The unbound ssDNA was removed from the NCM and incubated with a $His_6$-Rv0792c-immobilized membrane. After an hour, the membrane was washed with SB supplemented with 0.05% Tween 20 following an hour of incubation. The protein-bound aptamers were eluted by heating at 92°C and further enriched through PCR. The stringency of selection in successive rounds was enhanced by (i) increasing the number of washes (2 to 5 times), (ii) increasing the strength of buffer (0.05 to 1% Tween 20), (iii) decreasing the incubation time in the selection step, and (iv) increasing the incubation time for negative selection steps. The highest-affinity pool was cloned in the pTZ57R/T vector following six rounds of selection. The plasmid DNA was isolated from randomly picked transformants and confirmed by sequencing. Subsequently, the sequence similarity among the sequenced aptamers was performed using CLUSTALW and Bio Edit sequence alignment editors (https://www.nucleics.com/DNA_sequencing _support/Trace_viewer_reviews/BioEdit/) (70). The selected aptamers were screened using the Multiple Em for Motif Elicitation (MEME) online tool (http://meme-suite.org) to identify a conserved motif.

**ALISA and determination of $K_d$.** ALISA was performed as previously described (69). Briefly, a Nunc MaxiSorp 96-well plate was coated with 500 ng of purified wild-type or mutant proteins in 100 $\mu$L (0.1 mol/liter) sodium bicarbonate buffer (pH 9.6) at 37°C for 2 h. The plate was incubated with blocking buffer (5% bovine serum albumin and 0.25% Tween 20) at room temperature. After blocking, the wells were washed once with SB. This was followed by the addition of 100 pmol of 5′-biotinylated aptamer in SB, and plates were further incubated at room temperature. The wells were washed twice with SB supplemented with 1% Tween 20 and SB only after 1 h of incubation. Following this, 1:3,000 streptavidin-horseradish peroxidase (Strep-HRP) was added for 1 h at room temperature. The unbound Strep-HRP was removed by washing, and 100 $\mu$L of 3,3′,5,5′-tetramethylbenzidine was added. The reaction was stopped after 5 to 10 min of incubation by the addition of 100 $\mu$L of 5% $H_2SO_4$. The quantification of the protein-bound aptamer-Strep complex was determined by measuring absorbance at 450 nm. For $K_d$ determinations, 500 ng of protein was coated per well. After blocking, aptamer was added in the range of 2 to 500 nM, and ALISA was performed as described above. Next, the absorbance reading at 450 nm was plotted as a function of aptamer concentration, and the $K_d$ was calculated using a nonlinear regression model in GraphPad Prism.

**EMSAs for aptamer and promoter binding with Rv0792c.** For EMSAs, 100-pmol amounts of selected aptamers were incubated with 8 $\mu$M or 16 $\mu$M Rv0792c protein in binding buffer (10 mM Tris [pH 7.8], 10 mM $MgCl_2$, 50 mM KCl, 25 mM NaCl, 0.5 mg salmon sperm DNA, 0.75 mg BSA, 2.5% glycerol) for 20 min at 4°C. The reactions were resolved on 2% agarose gels, and the protein-aptamer complexes were detected by staining using SYBR safe. Promoter binding assays were performed using Cy5-labeled promoter fragment and 1 $\mu$M $His_6$-Rv0792c in binding buffer (10 mM Tris [pH 7.4], 50 mM NaCl, 10 mM $MgCl_2$, 0.2 mg/mL BSA, 10% glycerol, 1 mM dithiothreitol, 200 ng of sheared herring sperm DNA). Incubation was performed on ice for 20 min, and subsequently the reactions were resolved on 6% non-denaturing polyacrylamide gels and bands were visualized using a gel documentation system.

**SAXS and model building.** The SAXS studies were performed on an in-house SAXspace instrument using a line collimation source (X-ray wavelength of 0.15414 nm; Anton Paar, Graz, Austria). The samples were studied with sample-to-detector distance of 317.06 mm. Approximately 50-$\mu$L volumes of unliganded proteins and aptamers, their molar mixtures and matched buffer were transferred in a 1-mm

thermostatted quartz capillary and scattered X-rays were monitored on a one-dimensional Mythen detector. For further data processing, an average of three frames of 1 h were obtained. The programs used for different steps of data collection and processing are listed in Table S2a. The ATSAS suite of programs (v 2.8.4) was used to analyze SAXS data (71). Using shape constraints in the SAXS data profile, the shape of the predominant scattering particles was restored. Residue details of models for unliganded proteins and aptamers and their docked structures were compared and superimposed using automated alignment of inertial axes with CRYSOL and SUPCOMB programs, respectively, in the suite.

The models of unliganded His$_6$-Rv0792c and ssDNA aptamers with residue or base-level details were generated using the primary structures of protein and aptamers. The SWISS-MODEL server was used to search for structural templates of Rv0792c (https://swissmodel.expasy.org) (72). The results provided the putative template for Rv0792c from residues 53 to 286 of lin2111 from *Listeria innocua* [PDB ID 3EDP]. Using molecular dynamics analysis of the segments, the amino-terminal residues 1 to 52 and carboxy-terminal segment from residue 287 to 303 were generated. MD stimulation studies were performed using the Tinker molecular modeling package (v 4.2) along with an OPLSUA force field. The advanced Newton Raphson method was employed to compute structures of segments at 298K in implicit water ($\varepsilon = 80$). Simulations were run for 10 ns with restart coordinates written at every 1 ps. The predominant low-energy structures were filtered out as described previously (73, 74). SAXS data supported a dimeric state of unliganded Rv0792c; thus, by using the dimer as a central scaffold, two copies of predominant low-energy conformations of N- and C-terminal segments were aligned in space using the SASREF program, as reported previously (75). The composite structure of Rv0792c was generated and energy minimized by performing template-based modeling using the SWISS-MODEL server. The ELNEMO server was used to compute low-frequency collective vibrations accessible to the protein structure (76). ssDNA aptamers were modeled using their sequences and the ICM 3.8 program. SAXS data supported their masses to be close to their monomers, and thus monomeric forms of all three aptamers were considered for modeling studies. Repeated runs of global minimization and local optimization were performed until the conformations did not change more than 0.01 root mean square deviations (RMSD) across all atoms. By comparing theoretical SAXS profiles of the 10 lowest-energy conformations of aptamers with experimental SAXS data on the aptamers, the best conformation of aptamer agreeing with experimental data was identified. Further, the ELNEMO server was utilized to compute the most collective low-energy vibration mode to compare with SAXS data-based information and shape. These structures of ssDNA aptamers were docked on the structure of dimeric Rv0792c protein, and their pose on the protein was identified using SAXS data of the complexes as reference. The graphs pertaining to SAXS data analysis were prepared using OriginLab v5 software. The images of molecular models were prepared using the open source PyMOL v 1.1 and UCSF Chimera software v 1.14 programs.

***In silico* screening of drug-like molecules.** Molecular docking studies were performed using ICM Chemist Pro software v3.8. With the interaction distance mapping option, all residues within 3 Å of the interacting surface of Rv0792c (to aptamers) from the models of protein:aptamer complexes were selected. From both chains of Rv0792c, stretches of residues 125 to 137, 224 to 231, 253 to 257, 279 to 289, and 300 to 303 were selected to form the aptamer binding site. The library of approved drugs from drugbank.ca was used for docking studies, and the docking was done in an automated manner. Full degrees of freedom and rotations were given to the ligand during evaluation of docking poses on the identified receptor surface. The docking was carried out individually with each ligand, and its various poses with respect to the receptor pocket's charge and shape profile were calculated. The scores obtained for the docked pose were then arranged from low to high, and the top 10 lowest-scoring ligands were further selected for optimizing receptor residues around the low-score pose of ligand to obtain the new score. The short-listed ligands were further arranged according to the 4D docking score.

Next, we performed competitive ALISA to determine the ability of the top two hits to compete with the binding of aptamer to Rv0792c. The coating of the wild-type protein and blocking of nonspecific sites were performed as described above. Subsequently, the binding of aptamer was determined in the presence or absence of the top two small-molecule inhibitors. For IC$_{50}$ determination assays, inhibition assays were performed in the presence of 2.0-fold serial dilutions of small molecules. IC$_{50}$ values were calculated as the drug concentration that showed 50% inhibition for aptamer binding with Rv0792c.

**Intracellular killing experiments.** For macrophage experiments, THP-1 cells (human monocytic cell line) were differentiated into macrophages by the addition of 20 ng/mL phorbol 12-myristate 13-acetate (PMA). THP-1 macrophages were seeded at a density of $2 \times 10^5$ per well in a 24-well plate and infected with a single-cell suspension of *M. tuberculosis* at a multiplicity of infection of 1:10. The extracellular bacteria were removed after 4 h of infection by overlaying macrophages with RPMI containing 200 $\mu$g/mL amikacin. The infected macrophages were washed with 1× PBS twice, and macrophages were overlaid with drugs at noncytotoxic concentrations. The lysates were prepared by overlaying macrophages with 1× PBS containing 0.1% Triton X-100. For bacterial enumeration, the lysates were serially diluted and 100 $\mu$L was plated on MB7H11 medium at 37°C for 3 to 4 weeks.

**Statistical analysis.** GraphPad Prism 8 software (version 8.4.3; GraphPad Software Inc., CA, USA) was used for statistical analysis and graph generation. Significant differences between indicated groups were calculated using either a one-way analysis of variance (ANOVA) or the *t* test and were considered significant at a *P* value of <0.05.

**Data availability.** The raw data files for RNA-seq experiments have been deposited at NCBI and assigned accession number PRJNA727912. The data pertaining to unliganded protein, aptamer, and protein-aptamer complexes is available at https://www.sasbdb.org/project/1396/kv4wukfdsj.

## SUPPLEMENTAL MATERIAL

Supplemental material is available online only.

**SUPPLEMENTAL FILE 1**, XLSX file, 0.02 MB.
**SUPPLEMENTAL FILE 2**, PDF file, 1.7 MB.

## ACKNOWLEDGMENTS

R.S. acknowledges the funding received from Department of Biotechnology, India (Grant ID; BT/PR30215/MED/29/1343/2018). T.K.S. thanks Department of Biotechnology, India for funding support through Translational Research Program (BT/PR30159/MED/15/188/2018). Financial Support provided to N.K.C. from the DST SERB-NPDF Program (PDF/2016/002392) is gratefully acknowledged. A.S. acknowledges research fellowship received from Indian Council of Medical Research. T.P.G. is also thankful to Department of Biotechnology for her fellowship. Padam Singh acknowledges National Postdoctoral Fellowship received from Department of Science and Technology (PDF/2018/002454). G.C. acknowledges Ministry of Human Resource and Development for his research fellowship. We acknowledge Ashok Mukherjee for histopathology analysis. We are also thankful to Infection disease research facility and small animal house staff members at THSTI for technical help. R.S. is a recipient of Ramalingaswami fellowship and National Bioscience Award from Department of Biotechnology. R.S. is a senior fellow of Wellcome Trust-DBT India Alliance. We acknowledge lab attendants Rajesh and Sher Singh for technical help.

R.S. and T.K.S. conceived and designed the work plan. N.K.C., A.S., Padam Singh, and T.P.G. performed cloning, biochemical, microbiology and animal experiments. A.A. performed SELEX based experiments. K.D. and A. performed SAXS studies. E.K. and A.K. performed CD studies. P.S. performed AUC studies. R.S., T.K.S., A., D.S., and A.K. supervised the experiments performed in their respective laboratories. R.S., T.K.S., A., and N.K.C. analyzed the data, interpreted them, and wrote the paper as well.

We declare that the research was conducted in the absence of any commercial or financial relationships that could be considered as a potential conflict of interest.

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
