## [Reviewer comments · Microbiology Spectrum]

Microbiology Spectrum

Structural and functional characterization of Rv0792c from *Mycobacterium tuberculosis*: identifying small molecule inhibitors against HutC protein

Neeraj Chauhan, Anjali Anand, Arun Sharma, Kanika Dhiman, Tannu Gosain, Prashant Singh, Padam Singh, Eshan Khan, Gopinath Chattopadhyay, Amit Kumar, Deepak Sharma, Tarun Sharma, and Ramandeep Singh

Corresponding Author(s): Ramandeep Singh, THSTI

Review Timeline:

Submission Date:	May 27, 2022
Editorial Decision:	June 27, 2022
Revision Received:	September 25, 2022
Accepted:	October 1, 2022

Editor: Gyanu Lamichhane

Reviewer(s): Disclosure of reviewer identity is with reference to reviewer comments included in decision letter(s). The following individuals involved in review of your submission have agreed to reveal their identity: Sandhya S Visweswariah (Reviewer #2)

Transaction Report:

DOI: <https://doi.org/10.1128/spectrum.01973-22>

June 27, 2022

Dr. Ramandeep Singh
THSTI
Infection and Immunology Group
NCR Biotech Cluster
Faridabad, Haryana 121001
India

Re: Spectrum01973-22 (**Structural and functional characterization of Rv0792c from *Mycobacterium tuberculosis*: identifying small molecule inhibitors against HutC protein**)

Dear Dr. Ramandeep Singh:

Thank you for submitting your manuscript to Microbiology Spectrum. Two referees with high-level expertise in the topic of mycobacteriology have reviewed your manuscript and provided detailed critiques/recommendations.

Link Not Available

Sincerely,

Gyanu Lamichhane

Journals Department
Reviewer comments:

Reviewer #1 (Comments for the Author):

In this manuscript Chauhan et al. describe the functional characterization of Rv0792c, a HutC homolog from M.tb. A deletion mutant of Rv0792c showed reduced survival on exposure to oxidative stress, and in the guinea pig infection model, pointing to its importance in the pathophysiology of M.tb. Transcriptome analysis suggested that Rv0792c regulates the expression of genes stress adaptation and virulence, following which the authors used a combination of SELEX and SAXS data based

modelling to identify residues essential for the DNA binding activity of Rv0792c.

While the subject of this investigation is important in the context of M.tb pathogenesis, the following points may help to significantly improve the quality of the manuscript and the value of the data within:

54: The authors state that "To the best of our knowledge, this is the first detailed study for shape-function characterization of HutC-family of transcription factor from a bacterial pathogen". This assertion is only partially correct - an in-silico based structure analysis of Rv0792c, that included virtual inhibitor screens, was published last year - Abeywickrama, T.D.;Perera, I.C. In Silico Characterization and Virtual Screening of GntR/HutC Family Transcriptional Regulator MoyR: A Potential Monooxygenase Regulator in Mycobacterium tuberculosis. *Biology* 2021, 10, 1241. <https://doi.org/10.3390/biology10121241>. The authors are advised to refer to this manuscript and accordingly modify the description of their findings wherever relevant.

122: "We report that Rv0792c is an autoregulatory transcription factor" - This statement is baseless, no experimental evidence has been provided for either Rv0792c being a transcription factor, or for its autoregulatory role.

341: Rv0792c was observed to a predominantly dimeric protein using sedimentation velocity ultracentrifugation - can this observation be validated with Gel Filtration chromatography? What happens to the oligomeric state of the protein at higher salt concentrations? Were cross-linking studies attempted? Which of the detected oligomeric states of the protein are likely to be biologically relevant? How would one go about determining the oligomeric status of the protein in vivo?

354: Rv0792c is essential for the adaptation of M. tuberculosis upon exposure to oxidative 'stress' (missing word in quotes). Fig 3B is missing the complementation data for exposure to nitrosative stress, was the observed sensitivity restored when the mutant was complemented?

387: Complementation only partially restored the CFU counts in the lungs and spleens of infected guinea pigs. Could this be because Rv0792c is in all probability operonic with Rv0791c, Rv0790c and Rv0790c and the deletion has a polar effect on the expression of these genes? In such a scenario, full restoration of CFUs may require all the genes in the operon to be expressed in the deletion mutant. Testing the operonic status of Rv0792c would provide credence to this possibility.

412: Is the regulation of gene expression mediated by Rv0792c direct or indirect? Was the DNA binding activity of the protein tested with the 5'UTR regions of any of the DRGs? It might be pertinent to test this, as well as test the binding of the protein to the 5'UTR of its own gene. This approach may be more functionally appropriate compared to the extensive aptamer binding data that has been presented here to demonstrate the DNA binding activity of Rv0792c.

How does the constructed structural model of Rv0792c compare with the AlphaFold structure prediction of the protein?

The manuscript has multiple (too many to list) grammar, composition, and spelling mistakes (Fig 1C - 'Sedementation', being one example) - the authors are strongly urged to have the document professionally proofread.

Reviewer #2 (Comments for the Author):

The paper could do with major changes as indicated in the attachment. It could be made crisper and only relevant data shown as indicated in the review. Statistical analyses should be improved.

Staff Comments:

Preparing Revision Guidelines

- Point-by-point responses to the issues raised by the reviewers in a file named "Response to Reviewers," NOT IN YOUR COVER LETTER.
- Upload a compare copy of the manuscript (without figures) as a "Marked-Up Manuscript" file.

- Each figure must be uploaded as a separate file, and any multipanel figures must be assembled into one file.
- Manuscript: A .DOC version of the revised manuscript
- Figures: Editable, high-resolution, individual figure files are required at revision, TIFF or EPS files are preferred

Please return the manuscript within 60 days; if you cannot complete the modification within this time period, please contact me. If you do not wish to modify the manuscript and prefer to submit it to another journal, please notify me of your decision immediately so that the manuscript may be formally withdrawn from consideration by Microbiology Spectrum.

Chauhan *et al.* - Structural and functional characterization of Rv0792c from *Mycobacterium tuberculosis*: identifying small molecule inhibitors against HutC protein**Confidential remarks for the Editors**

In my opinion this manuscript is definitely not suitable for publication in its present form. It is poorly written and needs a proof-reading overhaul. More importantly, the design of the study is a bit dodgy - the authors should have performed a detailed functional characterization of Rv0792c for its DNA binding/ gene regulatory activity, as opposed to performing just the extensive aptamer based structural studies that have been presented here. They missed a trick by not testing the (highly likely) operonic status of Rv0792c, and incorporating this into their experimental design. Also, much of the structure prediction/ virtual inhibitor screening for the protein has already been published by Abeywickrama *et al.* (<https://doi.org/10.3390/biology10121241>), which takes away the novelty from some of the findings described.

I would've liked to recommend rejection, but the authors may be given an opportunity to submit a revision following the inclusion of additional experimental data based on the suggestions in my review.

Comments and Suggestions for the Authors

In this manuscript Chauhan *et al.* describe the functional characterization of Rv0792c, a HutC homolog from *M.tb*. A deletion mutant of Rv0792c showed reduced survival on exposure to oxidative stress, and in the guinea pig infection model, pointing to its importance in the pathophysiology of *M.tb*. Transcriptome analysis suggested that Rv0792c regulates the expression of genes stress adaptation and virulence, following which the authors used a combination of SELEX and SAXS data based modelling to identify residues essential for the DNA binding activity of Rv0792c.

While the subject of this investigation is important in the context of *M.tb* pathogenesis, the following points may help to significantly improve the quality of the manuscript and the value of the data within:

54: The authors state that “To the best of our knowledge, this is the first detailed study for shape-function characterization of HutC-family of transcription factor from a bacterial pathogen”. This assertion is only partially correct - an *in-silico* based structure analysis of Rv0792c, that included virtual inhibitor screens, was published last year - Abeywickrama, T.D.;Perera, I.C. *In Silico* Characterization and Virtual Screening of GntR/HutC Family Transcriptional Regulator MoyR: A Potential Monooxygenase Regulator in *Mycobacterium tuberculosis*. *Biology* 2021, 10, 1241. <https://doi.org/10.3390/biology10121241>. The authors are advised to refer to this manuscript and accordingly modify the description of their findings wherever relevant.

122: “We report that Rv0792c is an autoregulatory transcription factor” - This statement is baseless, no experimental evidence has been provided for either Rv0792c being a transcription factor, or for its autoregulatory role.

341: Rv0792c was observed to a predominantly dimeric protein using sedimentation velocity ultracentrifugation - can this observation be validated with Gel Filtration chromatography? What

happens to the oligomeric state of the protein at higher salt concentrations? Were cross-linking studies attempted? Which of the detected oligomeric states of the protein are likely to be biologically relevant? How would one go about determining the oligomeric status of the protein *in vivo*?

354: Rv0792c is essential for the adaptation of *M. tuberculosis* upon exposure to oxidative ‘stress’ (missing word in quotes). Fig 3B is missing the complementation data for exposure to nitrosative stress, was the observed sensitivity restored when the mutant was complemented?

387: Complementation only partially restored the CFU counts in the lungs and spleens of infected guinea pigs. Could this be because Rv0792c is in all probability operonic with Rv0791c, Rv0790c and Rv0790c and the deletion has a polar effect on the expression of these genes? In such a scenario, full restoration of CFUs may require all the genes in the operon to be expressed in the deletion mutant. Testing the operonic status of Rv0792c would provide credence to this possibility.

412: Is the regulation of gene expression mediated by Rv0792c direct or indirect? Was the DNA binding activity of the protein tested with the 5’UTR regions of any of the DRGs? It might be pertinent to test this, as well as test the binding of the protein to the 5’UTR of its own gene. This approach may be more functionally appropriate compared to the extensive aptamer binding data that has been presented here to demonstrate the DNA binding activity of Rv0792c.

How does the constructed structural model of Rv0792c compare with the AlphaFold structure prediction of the protein?

Did the authors consider testing the survival of the Rv0792 mutant in the THP-1 macrophage model of *M.tb* infection? From the data of the *in-vitro* stress experiments it is highly probable that one would observe a survival defect for the deletion mutant in this model. If this pans out, the effect of the identified inhibitor I-OMe-Tyrphostin can also be easily validated.

The manuscript has multiple (too many to list) grammar, composition, and spelling mistakes (Fig 1C - ‘Sedementation’, being one example) - the authors are strongly urged to have the document professionally proofread.

Response to Reviewers

We thank the reviewers for their careful assessment of our manuscript. We found their comments and suggestions useful to enable us to improve the manuscript. In the point-by-point response below the reviewers' comments are in bold and our responses are indented in *italics*. We sincerely hope that the revised manuscript would be considered suitable for publication in Microbiology Spectrum.

Reviewer's comments:

Reviewer #1:

The authors state that "To the best of our knowledge, this is the first detailed study for shape-function characterization of HutC-family of transcription factor from a bacterial pathogen". This assertion is only partially correct - an in-silico based structure analysis of Rv0792c, that included virtual inhibitor screens, was published last year - Abeywickrama, T.D.;Perera, I.C. In Silico Characterization and Virtual Screening of GntR/HutC Family Transcriptional Regulator MoyR: A Potential Monooxygenase Regulator in Mycobacterium tuberculosis. Biology 2021, 10, 1241. <https://doi.org/10.3390/biology10121241>. The authors are advised to refer to this manuscript and accordingly modify the description of their findings wherever relevant.

We thank the reviewer for careful assessment of the manuscript. As suggested, we have incorporated this reference in the manuscript and referred to the findings of this study in the manuscript.

"We report that Rv0792c is an autoregulatory transcription factor" - This statement is baseless, no experimental evidence has been provided for either Rv0792c being a transcription factor, or for its autoregulatory role.

We thank the reviewer for this valuable comment. In the revised manuscript, we have included data for EMSA showing that Rv0792c is able to bind to its own promoter. We have also included data showing that I-OMe-Tyrphostin inhibits the ability of Rv0792c to bind to its own promoter. This data has been included as Fig. 9 of the manuscript.

Rv0792c was observed to a predominantly dimeric protein using sedimentation velocity ultracentrifugation - can this observation be validated with Gel Filtration chromatography? What happens to the oligomeric state of the protein at higher salt concentrations? Were cross-linking studies attempted? Which of the detected oligomeric states of the protein are likely to be biologically relevant? How would one go about determining the oligomeric status of the protein in vivo?

As suggested, we have now performed SEC-MALS experiments. In the revised manuscript, we show that the major peak (peak 1) corresponds to a homodimeric form of the protein in SEC-MALS experiment. The observed molecular weight of the fraction was 68.4 kDa and the expected

molecular weight of the homodimer is 72.8 kDa. We also observed two additional peaks which most likely to higher order aggregates. We have included this data as Fig. 1C in the revised manuscript. In the present study, we have neither performed cross-linking experiment nor studied the oligomeric form of the protein in higher salt concentration. We speculate that the dimeric form is the biological relevant form of Rv0792c as reported in the case of other GntR homologs.

Rv0792c is essential for the adaptation of *M. tuberculosis* upon exposure to oxidative 'stress' (missing word in quotes). Fig 3B is missing the complementation data for exposure to nitrosative stress, was the observed sensitivity restored when the mutant was complemented?

We apologize for these errors. We have changed 'oxidative' to 'oxidative stress' in the revised manuscript. There was an error in uploading Fig. 2 in original manuscript. In the revised manuscript we have uploaded the correct Fig. 2. As mentioned in the original manuscript, we did not observe any defect in the survival of mutant strain upon exposure to nitrosative stress (Fig. 2B).

Complementation only partially restored the CFU counts in the lungs and spleens of infected guinea pigs. Could this be because Rv0792c is in all probability operonic with Rv0791c, Rv0790c and Rv0790c and the deletion has a polar effect on the expression of these genes? In such a scenario, full restoration of CFUs may require all the genes in the operon to be expressed in the deletion mutant. Testing the operonic status of Rv0792c would provide credence to this possibility.

As pointed out by the reviewer, we observed that complementation of the mutant strain partially restored the CFU counts in lungs and spleens of infected guinea pigs. In our RNA-seq data, we observed that the transcript levels of Rv0791c, Rv0790c were increased in the mutant strain. This observation suggests that there is a possibility of an alternative transcript start site present upstream of Rv0791c. We have included this statement in the discussion section of the manuscript. In subsequent studies, we will study the operonic status of Rv0792c locus.

Is the regulation of gene expression mediated by Rv0792c direct or indirect? Was the DNA binding activity of the protein tested with the 5'UTR regions of any of the DRGs? It might be pertinent to test this, as well as test the binding of the protein to the 5'UTR of its own gene. This approach may be more functionally appropriate compared to the extensive aptamer binding data that has been presented here to demonstrate the DNA binding activity of Rv0792c.

*We thank the reviewer for this valuable suggestion. As suggested by the reviewer, we have performed EMSA using Cy5 labelled promoter fragment. We show that Rv0792c is able to bind to its native promoter. Also, we noticed that preincubation of Rv0792c with I-O-Me-Tyrphostin reduced the promoter binding activity of Rv0792c in a concentration dependent manner. Further we have also performed intracellular killing experiments and show that I-O-Me-Tyrphostin was able to inhibit the growth of intracellular *M. tuberculosis*. We have included this data in Fig. 9 of the revised manuscript. We aim to perform CHIP-seq experiments in future to determine if the regulation of gene expression by Rv0792c for other DEGs mediated is direct or indirect.*

How does the constructed structural model of Rv0792c compare with the AlphaFold structure prediction of the protein?

AlphaFold and AlphaFold2 servers were used periodically. Both in Auto and multimer mode. All attempts failed due lack of usable templates in the database(s).

Upon receiving this review, a new search was again attempted using the primary structure. It failed. Please see the code crash report below.

Downloading alphafold2 weights to ..: 100%|██████████| 3.47G/3.47G [00:22<00:00, 169MB/s]

2022-07-09 23:01:14,140 Found 6 citations for tools or databases

2022-07-09 23:01:19,897 Query 1/1: gnr_b8154 (length 304)

PENDING: 0%| | 0/150 [elapsed: 00:00 remaining: ?]

2022-07-09 23:01:20,153 Sleeping for 5s. Reason: PENDING

COMPLETE: 100%|██████████| 150/150 [elapsed: 00:05 remaining: 00:00]

2022-07-09 23:01:25,653 Could not get MSA/templates for gnr_b8154: invalid literal for int() with base 10:

'MGSSHHHHHSSGLVPRGSHMASMTGGQQMGRGSMSTSVKLDLDAADLRISRGSPASTQL
AEALKAQIIQQRLPRGGRLPSERELIDRSGLSRVTVRAAVGMLQRQGWLVRRQGLGTFVADPV
EQELSCGVRTITEVLLSCGVTPQVDVLSHQTPAPQRISSETLGLVEVLCIRRRIRTGDQPLALVT
AYLPPGVGPAV

Traceback (most recent call last):

File "/usr/local/lib/python3.7/dist-packages/colabfold/batch.py", line 1336, in run

host_url,

File "/usr/local/lib/python3.7/dist-packages/colabfold/batch.py", line 821, in get_msa_and_templates

host_url=host_url,

File "/usr/local/lib/python3.7/dist-packages/colabfold/colabfold.py", line 229, in run_mmseqs2

M = int(line[1:].rstrip())

ValueError: invalid literal for int() with base 10:

'MGSSHHHHHSSGLVPRGSHMASMTGGQQMGRGSMSTSVKLDLDAADLRISRGSPASTQL
AEALKAQIIQQRLPRGGRLPSERELIDRSGLSRVTVRAAVGMLQRQGWLVRRQGLGTFVADPV
EQELSCGVRTITEVLLSCGVTPQVDVLSHQTPAPQRISSETLGLVEVLCIRRRIRTGDQPLALVT
AYLPPGVGPAV

2022-07-09 23:01:25,657 Done

The manuscript has multiple (too many to list) grammar, composition, and spelling mistakes (Fig 1C - 'Sedimentation', being one example) - the authors are strongly urged to have the document professionally proofread.

We apologize for this error. As suggested, we have carefully reviewed our manuscript for spell check and grammatical errors. We apologize for these errors and have rectified them in the revised manuscript. We have also had proofread our manuscript by a senior colleague in the institute.

Reviewer #2:

Much of the data could be shown as Supplemental data to make the paper crisper. For example, Figure 1B, the gel picture of induced and uninduced cultures need not be shown. The purified protein could be shown as a single image as an inset in the sedimentation data profile and there is no need to show different fractions following elution from the Ni-NTA column. In fact, perhaps the whole of Figure 1 could go be shown in the Supplementary information.

As suggested, we have removed Fig. 1B and included data for SEC-MALS experiment in Fig.1C of the manuscript.

In data where changes are not seen when grown under different conditions, all could go to Supp data. It appears that growth differences are half a log order different, so it would be better that actual cfu's are shown across independent experiments rather than a bar graph. There is also no comment made on why the growth defect is seen only by day 3.

In this study, we have compared the survival of wild type, mutant and complemented strain upon exposure to difference stress conditions. We observed that the mutant strain showed a growth defect upon exposure to oxidative stress only. We observed that the mutant strain displayed a growth defect of ~11.0- and 20.5-fold upon exposure to oxidative stress for 1 day and 3 days, respectively. The growth defect seen at day 3 was statistically significant. However, the differences observed at day 1 post exposure was not statistically significant. We are of the opinion, the data shown in Fig. 2 is important so we wish to include it as Main figure in the manuscript. As suggested by the reviewer, we have now shown actual cfu's from independent experiments in Fig. 2 of the revised manuscript.

In the RTPCR in Supp Figure S2, transcripts are seen for 0792. Is this because of the region where the primers were designed? Would it not have been better to use primers that lie in the region that was deleted? I also see that the complement has a tendency to show higher transcript levels? There is no comment made on this nor statistics shown in the graph. However, the complement could not restore completely the various phenotypes tested. In fact, in Figure 3, it is not clear which statistical groups are being evaluated. Would a one-way Anova be more appropriate than the t-test to check for significance?

In the present study, we have generated Rv0792c mutant strain of M. tuberculosis using temperature sensitive mycobacteriophages. The generation of mutant strain was confirmed by both PCR and qPCR using locus specific primers. The transcripts seen in the mutant strain were almost negligible (Fig. S2) and were 0.0001 times the levels seen in wild type strain. We have changed the scale of axis of this figure in the revised manuscript for better clarity. In the case of complemented strain, the transcript levels were ~ 2.4 folds the levels seen in the wild type strain. The increase in Rv0792c levels in the complemented strain obtained from two replicate experiments was not statistically significant. As suggested by the reviewer, we have not performed statistical analysis using one-way Anova and included this data in the revised manuscript.

Generally in all graphs with statistics, why is data shown as SEM and not SD? No indication of how many times experiments were repeated is stated anywhere. Are the results of infection studies shown from a single experiment? Again, the variation across different animals would be better to show rather than the bar graph.

As suggested by the reviewer, we have changed data to mean \pm S.D. in the revised manuscript. The results of infection studies are from a single experiment. In this experiment there were 6 animals per group per time point. As suggested, by the reviewer, we have now shown CFU obtained from different animals in Fig. 3 of the manuscript. We have also included the number of replicates in figure legends for each manuscript.

There is no mention of how many replicate cultures were sent for RNAseq analysis. The FDR values should be mentioned, not p-value for fold changes, which I presume is what is being shown. The resolution in Figure 4A is very poor-one cannot make out gene names. What do the green dots represent?

RNA-seq data was obtained from three replicate samples. We have now included this information in legend of Figure 4. As suggested, we have now included p-value and q-value for differentially expressed in Table S3 of the revised manuscript. Also, we have replotted Fig. 4A for better clarity. In the revised figure, upregulated and downregulated genes in the mutant strain have been shown using red and green dots, respectively. Black dots represent genes whose expression was not statistically significant between wild type and mutant strain.

Figure 5: Panel 1 not necessary-no enrichment beyond round 2? This could be shown as supplemental data or not shown at all.

As suggested, we have removed this panel from the revised manuscript.

What concentration of drugs was used to test for antibiotic sensitivity in Supp Fig S2F? Not mentioned. How long were cultures grown? Under what conditions?

For drug tolerance experiments, mid-log phase cultures ($OD_{600nm} \sim 1.0$) were exposed to different drugs such as isoniazid, rifampicin and levofloxacin at 10 $\mu\text{g/mL}$, 10 $\mu\text{g/mL}$ and 0.4 $\mu\text{g/mL}$, respectively. The strains were exposed to drugs for 14 days. We have now included these details in the methods section of the revised manuscript.

Not at all clear what sequence alignment in 5B means. They are rather rich in T's, and why do N's appear in the sequence alignment? It is not readily apparent as to what the sequence similarity is. Perhaps some dots or an asterisk below residues that are identical would be helpful. Or a sequence conservation logo would be even better.

As suggested, we have revised Fig. 5B. The new alignment figure is provided with dots. The dots show nucleotide conservation at that particular position. Also, we have created a sequence conservation logo using a Web-based tool MEME and data is presented as Table S4. The presence of 'N' in the sequence similarity data evince the presence of aptamer sequences having A/T/G/C in a particular position.

Importantly, are aptamer sequences seen in the Mtb genome? Are T rich sequences seen in the GC rich Mtb genome? Are these sequence motifs seen upstream of genes seen to be regulated in RNAseq? None of these issues are discussed in the paper.

We appreciate the concern raised by the reviewer. We would like to mention that Rv0792c specific aptamer sequences identified in the current manuscript are selected out from a commercially available synthetic DNA aptamer library not from the M. tuberculosis genome. In this study, SELEX approach was used as recognition sequence for Rv0792c are still unknown.

In Line 461-do the authors mean “sequence homology”? Or “similarity”? Is there any structural similarity in aptamer binding site in protein with that of GntR DNA binding regions?

We have modified sequence homology to sequence similarity in the revised manuscript. The SAXS data revealed that the screened aptamers were binding to C-terminal dimerizing segment of Rv0792c protein. In order to predict similarity between aptamer binding and DNA binding regions of Rv0792c, more experiments are required.

The legend to Figure 6A, B, C do not correspond to data shown. 6A shows mutant data not that with other proteins. B does not show dissociation data. C is not binding with different ligands.

We sincerely apologize for this error. We have uploaded the correct figures (Fig. 6A, 6B and 6C) along with the revised manuscript.

There are no statistics shown in 8C, D. A full dose-response curve is required for Tyrphostin. For example what inhibition is seen below 50µM concentrations? A word of caution about chasing tyrphostins as backbones for drug design. They are known to inhibit a number of diverse enzymes, and not just DNA binding proteins.

As suggested, we have now included statistics for Fig. 8C. We observed that mutation of either proline 40 or arginine 41 to alanine reduced the aptamer binding ability of Rv0792c. The reduction in activity of the mutant protein was found to be statistically significant. As shown in Fig. 8E, we observed that Tyrphostin inhibits the aptamer binding activity of Rv0792c by ~ 25%, 20% and 10% at 25 µM, 10 µM and 5 µM, respectively. We agree with the reviewer regarding chasing Tyrphostin as backbones for drug design. In the discussion section, we have mentioned that I-OMe-Tyrphostin and Tyrphostin has also been shown to inhibit Dop, a depupylase from M. tuberculosis. We have added that I-O-Me-Tyrphostin inhibited the DNA binding activity of

Rv0792c and also the growth of M. tuberculosis by ~ 4.5-fold in macrophages. We believe that this study paves the way for design of more potent small molecule inhibitors against HutC protein from M. tuberculosis.

The discussion seems a relisting of data from the paper without any further insight. It should be improved. Or Results and Discussion Sections could be combined.

As suggested, we have revised the discussion section of the manuscript for better clarity.

Minor Comments

Line 657: couldn't-better to write did not

As suggested, we have incorporated this change in the manuscript.

Line 653: didn't to did not

As suggested, we have incorporated this change in the manuscript.

Fig 4C w.r.t.???

As suggested, we have incorporated this change in the manuscript.

Line 426-transcript levels of PROTEINS??? Transcript of levels of genes encoding proteins....

As suggested, we have incorporated this change in the manuscript.

October 1, 2022

Dr. Ramandeep Singh
THSTI
Infection and Immunology Group
NCR Biotech Cluster
Faridabad, Haryana 121001
India

Re: Spectrum01973-22R1 (**Structural and functional characterization of Rv0792c from *Mycobacterium tuberculosis*: identifying small molecule inhibitors against HutC protein**)

Dear Dr. Ramandeep Singh:

Thank you for making good-faith effort to revise your manuscript as recommended by the expert reviewers.

Your manuscript has been accepted, and I am forwarding it to the ASM Journals Department for publication. You will be notified when your proofs are ready to be viewed.

Sincerely,

Gyanu Lamichhane
Editor, Microbiology Spectrum

Journals Department
Supplemental Material: Accept
Supplemental Material: Accept